# A dynamics based separation of deep and shallow stratospheric circulation branches

Rasul Baikhadzhaev[1], Felix Ploeger[1,2], Peter Preusse[1], Manfred Ern[1], and Thomas Birner[3,4]

[1]Institute of Climate and Energy Systems: Stratosphere (ICE-4), Forschungszentrum Jülich, Jülich, Germany.
[2]Institute for Atmospheric and Environmental Research, University of Wuppertal, Wuppertal, Germany.
[3]Meteorological Institute, Ludwig-Maximilians-Universität München, Munich, Germany.
[4]Institute of Atmospheric Physics, Deutsches Zentrum für Luft- und Raumfahrt, Oberpfaffenhofen, Germany.

**Correspondence:** Rasul Baikhadzhaev (r.baikhadzhaev@fz-juelich.de)

**Abstract.** The wave-driven Brewer-Dobson circulation plays a crucial role in determining the transport of trace gases and aerosols in the stratosphere. We examine the structure of the circulation based on reanalyses data (ERA5, ERA-Interim, MERRA2, JRA55) and the Transformed Eulerian Mean and downward control framework, aiming for a dynamical separation of different circulation branches in terms of outflow generated by wave driving. The results show the existence of different circulation regimes, with a deep circulation branch mainly driven by planetary waves with wavenumbers 1-3, and a shallow circulation branch mainly driven by smaller-scale waves with wavenumbers greater than 3. We propose a definition of the separation level between shallow and deep branches as the lowest level where outflow from planetary waves is larger than outflow from smaller-scale waves. We show that this level occurs at approximately 22km (43hPa) with a weak annual cycle. This climatological structure is robust in various reanalyses. The variability of the circulation in the deep branch above the separation level is mainly related to planetary waves, while the variability in the shallow branch is related to both smaller-scale and planetary waves. Trends in the circulation over the period 1980-2017 show an upward shift of the deep branch related to planetary waves and a downward shift of the shallow branch related to both planetary and smaller scale waves. The height of the separation level shows no significant trend. Taking into account differences in wave driving between the branches of the circulation could explain the spread in model inter-comparisons.

## 1 Introduction

The stratospheric meridional overturning circulation, also termed the Brewer–Dobson circulation (BDC), plays a crucial role in the climate system as it shapes the distributions of chemical and radiatively active species in the upper troposphere and lower stratosphere (UTLS). Particularly in the UTLS region, small changes in radiatively active species, like water vapour and ozone, can cause substantial radiative effects (e.g. Solomon et al., 2010; Riese et al., 2012) and impacts on atmospheric circulation (Charlesworth et al., 2023).

The BDC is characterized by upward transport of air masses in the tropical stratosphere, poleward transport in the stratosphere, and downward transport at middle and high latitudes (Holton et al., 1995; Butchart, 2014). As a mechanically–forced circulation, the BDC is driven by atmospheric waves travelling upwards from the troposphere. These waves dissipate throughout the

stratosphere and thereby decelerate the predominantly westerly background flow. This mechanism drives poleward mass flow

in the subtropical and middle latitude stratosphere and related upwelling in the tropics and downwelling at high latitudes. Hence, the global-scale Brewer-Dobson circulation is a wave-driven circulation in the meridional plane. Different waves drive the stratospheric circulation at different levels, including planetary-scale and synoptic-scale Rossby waves as well as small-scale gravity waves (e.g. Plumb, 2002). The structure of background flow in the stratosphere determines where the different waves dissipate, deposit their momentum, and finally cause the forcing of the circulation (Andrews et al., 1987).

From a zonal mean perspective, transport by the BDC can be separated into a residual mean mass circulation and additional, two-way eddy mixing without any net mass transport (Garny et al., 2014). Still, the two-way eddy mixing may cause transport of trace gases, due to the strong gradients of these gases in the stratosphere. The residual circulation velocities can be directly related to the wave forcing using the Transformed Eulerian Mean (TEM) framework (Andrews et al., 1987). As found by Haynes et al. (1991), the upward mass flow in the tropics is caused by the wave forcing from all atmospheric levels above

("Downward Control" principle). Hence, variations in tropical upward mass flow across a given level ("tropical upwelling") are expected to correlate with variations in the wave drag above that level.

A closer look at the detailed structure of the stratospheric circulation reveals different circulation branches (Plumb, 2002), comprising a shallow circulation branch in the lower stratosphere, and a deep circulation branch extending to high altitudes and latitudes (Birner and Bönisch, 2011). Birner and Bönisch (2011) found a clear distinction of these two circulation branches

in terms of transit times along the residual circulation and minimum pressure reached by the air parcels travelling with the residual circulation flow. Based on their trajectory analysis, the shallow circulation branch extends throughout the subtropics and mid-latitudes, vertically up to about 30–50 hPa. Motivated by these results, Lin et al. (2013) defined the shallow circulation branch to extend up to 30 hPa, the deep branch to be located above that level, and an additional transition branch in the lowest stratosphere below 70 hPa. These simplified definitions have the advantage that they equally divide the tropical upward mass

flux between the shallow and deep branches, and have been applied for multi-model inter-comparisons of the BDC strength and changes therein (Lin et al., 2013). Furthermore, the shallow and deep stratospheric circulation branches have been argued to be driven by different mechanisms. Gerber (2012) showed from simulations with a mechanistic model that the shallow branch is mainly "tropospherically controlled" by variations of wave sources close to the surface, whereas the deep branch is "stratospherically controlled" by the structure of the stratospheric wave guide.

In a future climate, the stratospheric circulation is expected to accelerate as a response to increasing carbon dioxide levels, and climate models simulate this increase very robustly and throughout the stratosphere (e.g. Butchart, 2014). As the stratospheric residual circulation is too slow to be directly measurable, its strength needs to be inferred indirectly from observations. One commonly used diagnostic for that is stratospheric mean age of air, i.e. the average transit time of air parcels while transported through the stratosphere (Waugh and Hall, 2002). In the lower stratosphere, most recent inter-comparisons show

decreasing stratospheric mean age of air, indicating a circulation increase over time also in observation-based datasets, and hence consistency with climate model simulations (Garny et al., 2024). At upper levels in the middle stratosphere, there is a remaining discrepancy between stratospheric circulation trends simulated by free-running climate models and observation-based estimates (Garny et al., 2024), which had been already noted for different generations of climate models (e.g. Abalos

et al., 2021; Butchart, 2014; Waugh, 2009). At higher altitudes, climate models typically simulate an accelerating circulation, while both in-situ and satellite observations show no evidence of such a trend (Ray et al., 2014; Mahieu et al., 2014; Stiller et al., 2017). However, related observational uncertainties are too large to disprove model predictions (Garny et al., 2024).

Meteorological reanalyses combine model simulations with observational data by data assimilation, and investigations of the stratospheric circulation in these datasets can give additional insights. In the lower stratosphere, most recent reanalyses show decreasing mean age while in the middle stratosphere mean age trends may differ even in sign between different reanalyses (Chabrillat et al., 2018; Ploeger et al., 2019). Such differences between trends in residual circulation and trends in age of air are likely related to the fact that age of air is not solely controlled by the residual circulation but also strongly affected by mixing processes (Waugh and Hall, 2002). Overall, differences in the vertical structure of stratospheric circulation trends between different datasets raise the question as to whether the dynamical processes driving these changes also differ vertically.

Analysis of the forcing of circulation trends in climate models at a given level shows that particularly the partitioning into contributions of different wave types differs substantially between different models (Abalos et al., 2021). These discrepancies in circulation and wave forcing trends might potentially be attributed to variations in the depths of circulation branches across different datasets, such that comparisons at a given level might contrast different dynamical regimes. Hence, an improved interpretation of the stratospheric circulation and its potential changes requires an improved and more detailed understanding how the different (shallow and deep) circulation branches are separated dynamically.

In this paper, we investigate the stratospheric residual mean mass circulation and its wave forcing and aim for a dynamical separation criterion for different circulation branches. The analysis involves circulation estimates based on the TEM framework and on the downward control principle, as well as separation into contributions from different waves. These diagnostics are applied to various meteorological reanalyses. Specific research questions raised for this paper are: (i) Can the established shallow and deep branches of the stratospheric circulation be dynamically defined based on the wave driving classified by spatial scale? (ii) Did different branches of the stratospheric circulation change differently over the past decades? (iii) How robust is the representation of the circulation structure and its changes in different reanalyses?

The applied circulation diagnostics and considered reanalysis data are described in Sect. 2. Thereafter, Sect. 3 investigates the climatological structure of the stratospheric circulation. Section 4 further focuses on variability and trends in the stratospheric circulation. The results are discussed in the context of existing literature in Sect. 5 before presenting the final conclusions of the paper.

## 2 Data and methods

### 2.1 Reanalysis data

Reanalyses aim to produce time-consistent datasets by integrating available observations through data assimilation to a 'fixed' version of Numerical Weather Prediction (NWP) model, thus providing the most complete data on past climate. The NWP model, its version, set of incorporated observations, and data assimilation methods depend on the assimilation center and on the version of the analysis. In this work we use data from four global reanalyses: the Modern-Era Retrospective analysis for

Research and Applications, Version 2 (MERRA2) from the National Aeronautics and Space Administration (NASA) (Gelaro et al., 2017), the Japanese 55-year Reanalysis (JRA55) from the Japan Meteorological Agency (JMA) (Kobayashi et al., 2015), the ERA-Interim (Dee et al., 2011) and ERA5 (Hersbach et al., 2020) reanalyses from the European Centre for Medium-Range Weather Forecasts (ECMWF). The ERA5 (European Centre for Medium-Range Weather forecasts Reanalysis v5) is the fifth generation ECMWF atmospheric reanalysis, covering the period from 1940 to present. The reanalysis has 137 vertical levels extending from the surface to 0.01 hPa. In ERA5, the stratosphere is represented by approximately 45 levels. In all four reanalyses vertical resolution gets gradually coarser with altitude; in ERA5, the resolution is about 350 m in the lowermost stratosphere and 1500 m in the uppermost stratosphere. The ERA5 high-resolution data is produced as spectral coefficients with a triangular truncation of T639 which roughly corresponds to horizontal resolution of 0.28°; the data is available at hourly intervals.

The ERA-Interim (Interim European Centre for Medium-Range Weather forecasts Reanalysis) is a predecessor of ERA5, it was discontinued in 2019 and covers the period from 1st January 1979 to 31st August 2019. ERA-Interim has 60 vertical levels spanning from the surface to 0.1 hPa. Approximately 20 of these levels are in the stratosphere, with a vertical resolution of about 1000 m in the lowest part of the stratosphere and roughly 3000 m at the top of the stratosphere. The spectral resolution of the reanalysis is T255, it corresponds to horizontal resolution close to 0.70°; the data is available at six-hour intervals.

The JRA55 data is available from 1958 and was discontinued at the end of January 2024, the topmost level of JRA55 is 0.1 hPa, it also has 60 vertical levels about 20 of which are in the stratosphere. Vertical resolution in stratosphere is similar to ERA-Interim. With spectral resolution of T319 the horizontal resolution of JRA55 is approximately twice as coarse as ERA5, at about 0.56°; the temporal resolution of the archived data is 6 hours.

The MERRA2 reanalysis covers the period from 1980 to present. The reanalysis has 72 levels spanning from the surface to 0.01 hPa, about 25 of the levels are in the stratosphere. Vertical resolution of the reanalysis in the lowermost stratosphere is close to 1000 m and to 2000 m at the top of the stratosphere. Native horizontal resolution of MERRA2 is 0.5° lat x 0.625° lon. Three-dimensional products are available at 3-hourly intervals.

The main focus is on ERA-5 which is the most recent reanalysis among these four. The enhanced horizontal and vertical resolution of ERA5 compared to the other reanalyses is expected to result in improved ability to resolve smaller-scale waves (Watanabe et al., 2015; Müller et al., 2018). Other reanalyses are mostly used for comparison and to assess the robustness of the results..

To retain consistency of spatial and temporal resolution in the inter-comparison, all used reanalyses were resampled on a 1x1 degree latitude-longitude grid, interpolated vertically to the same 88 pressure levels and resampled in time to 6 hours (00, 06, 12, 18 UTC). The comparison between different reanalyses has been done on the common 38 year period from 1980–2017. The effect of resampling on ERA5 data is further explored in Sect. 5.

The pattern of the residual circulation in the stratosphere is shown in Fig. 1 based on ERA5 reanalysis (purple arrows in left panel). The figure indicates evidence for the two main branches of the overturning circulation: the deep branch transfers mass from the tropical tropopause to the middle and upper stratosphere and towards the winter pole, and the shallow branch acting in the lower part of the stratosphere, creates poleward mass flux in both hemispheres. .

## 2.2 Residual circulation diagnosis

The analyses in this paper are based on the Transformed Eulerian Mean (TEM) framework (Andrews et al., 1987) utilizing the two independent estimates to diagnose and investigate the overturning circulation: (1) the "direct" residual circulation definition $\bar{v}^*, \bar{w}^*$ and (2) an "indirect" momentum balance-based estimate of the residual circulation $\bar{v}_D^*, \bar{w}_D^*$, where the upwelling velocity is calculated from the wave drag based on the Downward Control Principle (DWCP) (e.g. Haynes et al., 1991). Although both approaches are based on the TEM framework, we will refer to the direct estimate of the residual circulation as the "TEM" approach and to the indirect momentum balance estimate as the "DWCP" approach, in the following. On the one hand, the direct TEM approach is conceptually simpler and more frequently used in studies of the stratospheric circulation. On the other hand, the DWCP approach allows to relate the residual circulation to the wave drag and in particular to estimate the relative contributions of individual waves to the driving of the residual circulation. Uncertainties and limitations such as those arising from numerical approximations, data assimilation procedures, and the lack of strict mass conservation in reanalyses affect both estimates in distinct way. Therefore, their comparison provides valuable insight into the robustness of our results.

The vertical and horizontal components of the TEM residual circulation in log-pressure coordinates are defined through

$$
\bar{v}^* = \bar{v} - \rho_0^{-1}(\rho_0 \overline{v'\theta'}/\bar{\theta}_z)_z
$$
$$
\bar{w}^* = \bar{w} + (a\cos(\phi))^{-1}(\cos(\phi)\,\overline{v'\theta'}/\bar{\theta}_z)_\phi
\tag{1}
$$

where an overbar indicates the zonal mean, a prime the deviation from the mean, subscripts denote partial derivative, $v$ is meridional velocity, $\rho_0$ is basic density, $\theta$ is potential temperature, $w$ is vertical velocity, and $a$ is the mean radius of the Earth.

Energy deposition from wave dissipation constitutes the driving force of the residual circulation. The wave dissipation results in eddy potential vorticity fluxes which, in turn, are approximately equal to the Eliassen-Palm flux (EP flux) (Andrews et al., 1987). The latitudinal and vertical components of the EP flux are defined by

$$
F^{(\phi)} = \rho_0 a\cos(\phi)\left(\bar{u}_z \overline{v'\theta'}/\bar{\theta}_z - \overline{v'u'}\right)
$$
$$
F^{(z)} = \rho_0 a\cos(\phi)\{[f - (a\cos(\phi))^{-1}(\bar{u}\cos(\phi))_\phi]\overline{v'\theta'}/\bar{\theta}_z - \overline{w'u'}\}
\tag{2}
$$

where $u$ is zonal wind velocity, $f$ is the Coriolis parameter defined as $2\Omega\sin(\phi)$, where $\Omega$ is the angular velocity of the Earth. The divergence of the EP flux ($\nabla \cdot F$) represents the driving force in the zonal momentum balance equation

$$
\frac{\partial \bar{u}}{\partial t} + \bar{v}^*\left(\frac{1}{a\cos(\phi)v}\frac{\partial \bar{u}\cos(\phi)}{\partial \phi} - f\right) + \bar{w}^*\frac{\partial \bar{u}}{\partial z} = \frac{\nabla \cdot F}{\rho_0 a\cos\phi} + \bar{X}\,.
\tag{3}
$$

Here, $X$ includes contributions from parameterized waves and will not be further considered in this paper which focuses on the contribution from resolved waves.

The contributions of individual waves to the EP flux can be estimated by Fourier transformation of the three-dimensional EP flux data from longitude to wavenumber-frequency domain. The thus transformed components of the EP flux are defined through

$$
F^{(\phi)}(s) = \rho_0 a\cos(\phi)Re[\bar{u}_z\hat{v}(s)\hat{\theta}^*(s)/\bar{\theta}_z - \hat{u}(s)\hat{v}^*(s)]/(n_s^2/2)
$$
$$
F^{(z)}(s) = \rho_0 a\cos(\phi)Re\{[f - (a\cos(\phi))^{-1}(\bar{u}\cos(\phi))_\phi]\hat{v}(s)\hat{\theta}^*(s)/\bar{\theta}_z - \hat{u}(s)\hat{w}^*(s)\}/(n_s^2/2)
\tag{4}
$$

where $Re$ denotes the real part of a complex number, asterisks denote complex conjugates, hats represent Fourier coefficients, $s$ is the wavenumber index, and $n_s$ is the sample size (number of longitudinal grid points). To compute these terms, a Fast Fourier Transform (FFT) was applied to each fluctuation component, excluding the negative frequencies. This FFT results in a N/2 long set of individual wave contributions to the EP flux. Summation over all wave components in $F(s)$ from Eq. 4 yields values that are almost identical to those of $F$ in Eq. 2, such that numerical uncertainties in the calculation are negligible.

Here, wavenumber represents the spatial scale of the wave, the number of waves per 360 degrees of longitude, with wavenumber 1 (hereafter referred to as wave 1) being the largest possible zonal wavelength. In this work, the reanalysis data has been resampled to a 1 by 1 degree horizontal grid. The Nyquist-Shannon sampling theorem (Nyquist, 1928; Shannon, 1949) requires to have at least two data points per wavenumber to detect the wavenumber. Therefore we truncate the wave separation at wave 180, noting that even for this wavenumber the contribution could be underestimated (eg. Abdalla et al., 2013; Skamarock, 2004). The effect of resampling ERA5 from 0.3° to 1° horizontal resolution on the contribution of smaller-scale waves is discussed in more detail in Sect. 5

The wave dissipation given by the $\nabla \cdot F$ and the residual circulation are connected through the DWCP. The DWCP states that the mass flow across a given level is related to the momentum deposition from dissipating waves above that level. We use this relation to estimate the relative contributions of different waves to the driving of the residual circulation. Based on the DWCP, the following expressions can be derived for the residual mass stream function $\bar{\psi}_D^*$ and for the components of the residual circulation $\bar{v}_D^*, \bar{w}_D^*$

$$\bar{\psi}_D^* = \int_z^\infty \frac{a\cos(\phi)\nabla \cdot F}{[a\cos(\phi)(\bar{u} + a\cos(\phi)\,\Omega)]_\phi}\, dz \tag{5a}$$

$$\bar{v}_D^* = \frac{1}{\rho_0\cos(\phi)}\frac{\partial\bar{\psi}_D^*}{\partial z} \tag{5b}$$

$$\bar{w}_D^* = \frac{1}{a\rho_0\cos(\phi)}\frac{\partial\bar{\psi}_D^*}{\partial\phi} \tag{5c}$$

Strictly speaking, the integration in Eq. 5a should be performed along isolines of constant angular momentum. However, since outside the tropics the isolines of constant angular momentum are approximately lines of constant latitude, we perform the integration along constant latitudes (cf. Abalos et al., 2015). Therefore, we estimate the mean residual circulation upwelling between 20°S and 20°N by integration of Eq. 5c

$$\bar{w}_D^{*\pm} = \frac{\bar{\psi}_D^{*-} - \bar{\psi}_D^{*+}}{-a\rho_0 S^\pm} \tag{6}$$

where the superscripts $\pm$ denotes that the quantity is evaluated at the latitudes 20°N and 20°S, respectively. Hence, $S^\pm$ is the normalized area fraction of the latitude band between these latitudes and $\bar{w}_D^{*\pm}$ is the mean DWCP vertical velocity averaged over this region.

The turn-around latitudes (TAL) separate the area of upward residual velocity in the tropical stratosphere from the area of downward velocity in the extratropics (eg. Abalos et al., 2015). In this work, tropical upwelling is defined as the total mass flow between the TAL carried by $\bar{w}^*$, and tropical outflow is defined as the poleward mass flow across the TAL. The outflow

at a given level represents the result of "gyroscopic pumping" (Holton et al., 1995), indicating the amount of air mass being pulled from the tropics to middle and high latitudes by the wave drag at that level. The tropical upwelling, on the other hand, represents the integrated effect of the wave drag at all levels above, as described by the downward control principle (see above). Hence, the total tropical upwelling is given by

$$W = 2\pi a^2 \rho_0 \int\limits_{\phi_-}^{\phi_+} \bar{w}^* cos(\phi) d\phi \, , \tag{7}$$

and the outflow out of the tropics at a given level is the vertical derivative (top to bottom) of the upwelling

$$V = -\frac{\partial W}{\partial z} \, . \tag{8}$$

The two major wave types which drive the overturning circulation are Rossby and gravity waves. Planetary and synoptic scale Rossby waves are large-scale, and are related to the horizontal gradient of potential vorticity. These waves are influenced by the Earth's rotation, the latitudinal gradient of the Coriolis parameter, and the mean zonal flow. The gravity waves are of smaller-scale, less dependent on the Earth's rotation and primarily result from the restoring force due to stable background stratification. The upward propagation of Rossby waves is controlled by the Charney-Drazin condition (Vallis, 2006), which for stationary waves is

$$0 < \bar{u} < \frac{f_\phi}{k^2 + l^2 + (f_0/2NH)^2} \, . \tag{9}$$

Here, $k$ and $l$ are zonal and meridional wavenumbers, $f_0$ is a reference value of the Coriolis parameter, and $N$ is the buoyancy frequency. This condition implies that Rossby waves may propagate upwards into the stratosphere if the zonal flow is eastward and weaker than a critical value. This critical value is higher for larger-scale waves (smaller $k$, $l$) such that these waves may propagate deeper into the stratosphere compared to smaller-scale waves.

A common approach in BDC studies is a separation of wave-driving based on the zonal wavenumber of the involved atmospheric waves (eg. Randel et al., 2008; Kim and Alexander, 2015; Abalos et al., 2024). In this study, the following scale distinction is adopted: waves 1 to 3 are categorized as planetary, waves 4 to 20 as synoptic scale, and resolved waves with wavenumbers greater than 20 are classified as mesoscale waves. A combination of synoptic- and mesoscale waves will be referred to as smaller-scale waves. Accordingly, planetary waves are Rossby waves, synoptic scale waves are dominated by Rossby waves with increasing secondary contributions due to gravity waves for higher wavenumbers., and mesoscale waves can be interpreted as gravity waves (Strube et al., 2020; Žagar et al., 2018). Furthermore, planetary waves are typically forced by large-scale stationary features of the Earth's surface, such as orography and land-sea contrasts, and therefore at the altitude range of interest are expected to be stationary or quasi-stationary (Ghinassi et al., 2022). This interpretation is further supported by the consistency between panel b of Fig. 1 and the amplitudes of geopotential height variations associated with zonal wavenumber 1 in (Koval et al., 2023). In particular, the slowest normal mode with a 16-day period analysed by (Koval et al., 2023) most closely resembles the structure of the Eliassen–Palm flux shown in panel b.

To address the research questions outlined in the Sect. 1, a Fourier transformation is applied to 6-hourly snapshots of the fluctuation components, and $\nabla \cdot F(s)$ (with the zonal wavenumber domain replacing longitude domain) is calculated (see Eq.

4). After that, the monthly mean is computed from the $\nabla \cdot F(s)$ snapshots. Utilizing the DWCP framework (Eq. 5), the monthly
mean $\nabla \cdot F(s)$ related to specific wavenumbers is used to reconstruct the circulation driven by individual waves or sets of waves.
Subsequently, the associated upwelling and outflow are computed from the reconstructed circulation using Eq. 7, and 8. The
analysis is then conducted on the resulting upwelling and outflow fields.

## 3 Wave forcing of the Stratospheric circulation branches and separation level between them

### 3.1 General structure of the circulation

Figure 1 shows the climatology of the residual mean mass circulation for the period 1980-2017 together with the resolved wave
drag quantified in terms of $\nabla \cdot F$, which constitutes the circulation's driving force (see Eq. 6). Note that $\nabla \cdot F$ has been scaled by
$(a\rho_0 cos\phi)^{-1}$ and $86400$ s/day, to obtain the wave drag in $\mathrm{m\,s^{-1}\,day^{-1}}$. The well-known characteristics of the stratospheric
BDC, with upwelling in the tropics, poleward motion and downwelling in the extratropics, and a stronger circulation cell in
the winter hemisphere, are all evident in Fig. 1 (colors represent dissipation or "wave drag" by resolved waves).

The reduced wave drag in the middle and upper stratosphere in the summer hemisphere is related to the presence of westward
winds which prohibit Rossby waves to propagate upwards, known as the Charney-Drazin criterion (Charney and Drazin, 1961).
The wave filtering resulting from this criterion prohibits the propagation of waves through a medium with an eastward wind
speed above a certain threshold or any westward wind (see Eq. 9). Consequently, as illustrated in the second and third columns
of the figure, large-scale Rossby waves are more likely to satisfy the criterion and can generally propagate deeper into the
235 stratosphere. Furthermore, the majority of small-scale Rossby waves with wavenumbers larger than 3 dissipate in the lower
part of the stratosphere. Gravity waves remain unaffected by the filtering process, allowing them to propagate deeper into the
atmosphere and to drive the mesospheric circulation. The seasonality of the BDC causes seasonal movement of the TAL with
significant displacement into the summer hemisphere (Fig 1, dashed black lines).

The primary factor driving the tropical upwelling in the stratosphere is the wave drag near the TAL, as it is ultimately
responsible for the uplift in the tropics (Eluszkiewicz et al., 2000).

The northern TAL in the lower stratosphere is typically located between $20°$ to $40°$ N. Figure 2 presents the time series of
the wave drag and the horizontal component of the residual circulation, estimated using the TEM approach and DWCP, at 70
hPa within this latitude range. The figure shows a close relation between horizontal velocity $\bar{v}^*$ and wave drag, as expected
from Eq. 6. Overall, the net outflow velocity calculated using the TEM approach agrees well with the momentum balance
estimate of the velocity derived from the drag of all resolved waves. The discrepancy in residual circulation velocity between
the TEM approach (green dashed line in Fig. 2) and the DWCP (black dashed line) at 70 hPa is at lest in part attributable
to the fact that the DWCP only accounts for resolved wave drag (see Eq. 5a) while the full horizontal velocity based on the
TEM definition is also influenced by small-scale parameterized gravity waves. Other potential contributors to the discrepancy
include computational approximations and the known limitation that reanalyses do not strictly conserve mass. An effect of
250 parameterized gravity wave drag on upwelling similar to the discrepancy between the approaches has been shown by (Butchart
et al., 2011). As a result, the DWCP estimate of the residual circulation driven by the drag of all resolved waves is weaker than

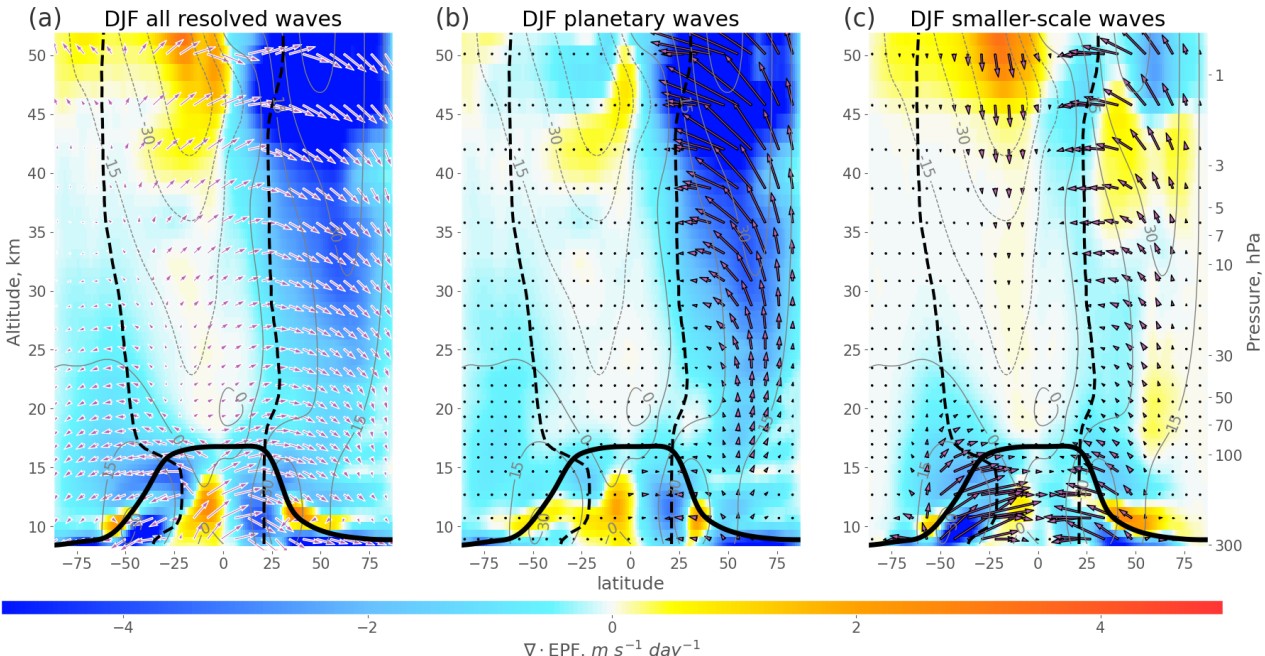

**Figure 1.** Climatological (1980-2017) distributions of resolved wave drag (estimated as $\nabla \cdot F$, color coded) together with Eliassen–Palm flux vectors (black contour arrows in panels b and c), log scaled TEM residual circulation (white contour arrows in panel a), zonal mean zonal wind (grey contours) from ERA5 reanalysis for December–February. The distributions are shown for the contribution from all resolved waves (left), from planetary waves (wavenumbers 1-3, middle) and from smaller-scale waves (wavenumber higher than 3, right). The black dashed lines show the turn-around latitudes (where upwelling in tropics changes to downwelling in extratropics), the thick black line shows the tropopause.

the one obtained taking into account also the additional wave drag related to parameterizations, using the TEM approach. This deficit is shown in Fig. A2 and discussed in more detail in Sect. 5.

As might be expected, the wave drag at the considered level $\nabla \cdot F$ (solid lines in Fig. 2), and the horizontal component of the residual circulation estimated from the wave drag above using the DWCP $\bar{v}_D^*$ (dashed lines in the figure) are consistent and have similar variability. The wave driving at 70 hPa in the Fig. 2 shows that about 3/4 of the circulation at the level is being driven by smaller-scale waves. However despite driving only 1/4 of the circulation, the planetary waves influence the variability of the circulation almost as much as the smaller-scale waves. The variability of the circulation velocities and relation to wave drag are discussed in more detail in Sect. 4.1.

The wave filtering due to the Charney-Drazin criterion related to the pattern of zonal wind (Fig. 1) limits the propagation of smaller-scale Rossby waves into the middle and upper stratosphere. As a result, the smaller-scale waves that do reach the

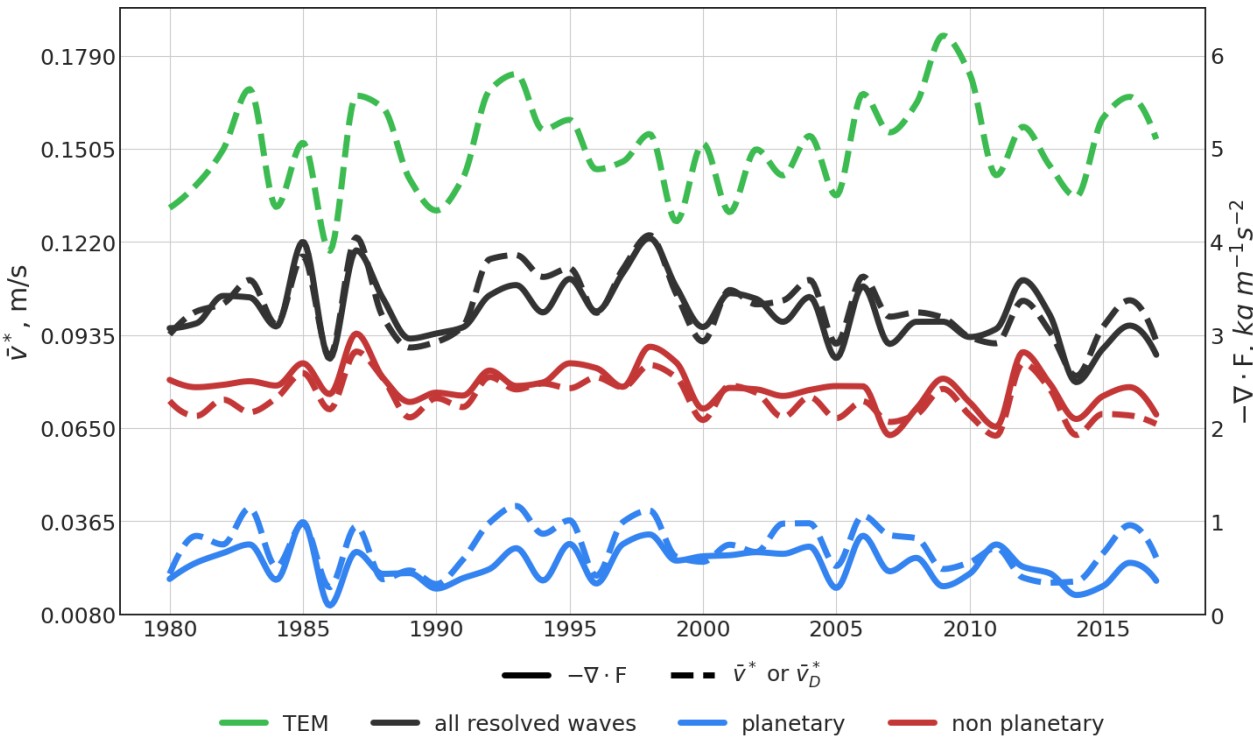

**Figure 2.** Annual mean time series at 70 hPa (mean 20N-40N) of horizontal component of the residual circulation (dashed lines), and $\nabla \cdot F$ flux (solid lines). Residual velocity from TEM approach (green), $\nabla \cdot F$ and residual velocities from DWCP for total (black), planetary (blue) and other than planetary (red) waves. The figure is based on 1980-2017 ERA5 data

stratosphere are expected to break and deposit their momentum at lower levels. Thus smaller-scale waves are expected to drive the shallow branch and planetary waves to drive the deep branch of the residual circulation (Plumb, 2002).

Analysis of the resolved wave drag demonstrates that the large-scale waves (planetary and synoptic scale) generate most of the upwelling in the tropical lower stratosphere (Fig. 3a). The separation of horizontal outflow into contributions from different waves in Fig. 3b further shows that at upper levels planetary waves dominate while at lower levels (below about 22km) synoptic-scale waves dominate. This indicates that the deep branch is mainly a result of planetary waves drag, while the shallow branch is mostly driven by synoptic scale waves. Furthermore, drag due to mesoscale waves almost exclusively contributes to the shallow branch. This drag induced by the breaking of mesoscale waves is significantly stronger in ERA5 than in the other 3 reanalyses (not shown).

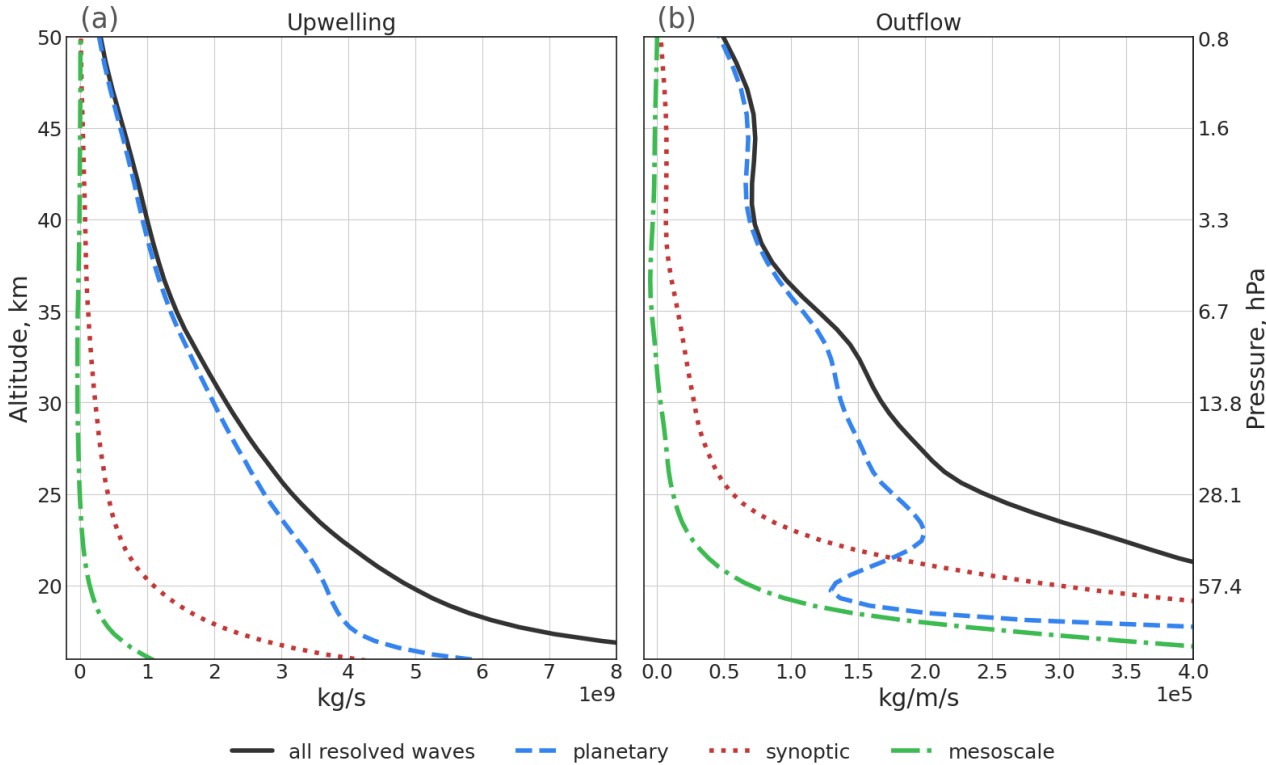

**Figure 3.** (a) Vertical profiles of mean (1980-2017) upwelling estimated with DWCP and driven by all resolved wave drag (black solid line), planetary wave drag (blue dashed line), synoptic scale wave drag (red dotted line), and mesoscale wave drag (green dash-dotted line). (b) equivalent to (a) with vertical profiles of mean outflow. The figure is based on ERA5 data

## 3.2 Separation of the deep and shallow branches by wave forcing based on climatological data

We conduct a more detailed analysis of the specific wave driving of the deep and shallow circulation branches based on both a mechanistic and a statistical analysis. The mechanistic analysis is based on the DWCP and separates upwelling and outflow velocities into contributions of different waves. Since planetary and synoptic scale waves account for the majority of the resolved wave drag, the contributions of the individual wavenumbers from 1 to 6 are separated in Fig. 4a. The vertical profiles of outflow presented in the figure clearly indicate that waves 1 and 2 drive the deep branch of the circulation at levels above about 22 km. On the other hand, wave 4 and smaller-scale waves dominate the upwelling below that level. The role of waves with wavenumber 3 and 4 is somewhat intermediate, with these waves driving a significant portion of the circulation at all levels. The other 3 reanalyses considered in this study exhibit similar vertical distributions of wave drag for the wavenumbers 1 to 6 (not shown). Regarding the relative outflow in Fig. 4b, waves with wavenumbers larger than 3 produce outflow maxima which are well-confined below about 25km, while the outflow due to larger-scale planetary waves maximizes over broad

regions above. The vertical location of the maxima of relative outflow indicates that wave 3 plays a more prominent role in driving the deep branch, while wave 4 is more strongly driving the shallow branch. The individual outflow induced by a single wavenumber up to wave 59 can be seen in Fig. A1, which shows that, resolved waves with scales smaller than wave 6 are almost exclusively driving the shallow branch of the stratospheric circulation.

The statistical analysis examines the correlation near the TAL between the outflow velocity and the $\nabla \cdot F$ associated with different wave bands. Here, the outflow velocity is estimated as the horizontal component of the residual circulation from the TEM framework ($\bar{v}^*$) (multiplied with minus sign in the northern hemisphere, such that a positive correlation results for increased poleward flow related to more negative $\nabla \cdot F$ values). The correlation coefficient between the TEM outflow velocity and the $\nabla \cdot F$, derived from the total resolved wave drag, is approximately 0.8 to 0.9 — except in regions below 20 km altitude, where the correlation weakens (not shown). To further investigate whether waves 3 and 4 primarily drive the shallow or deep branch, the correlations of both large-scale wave drag with outflow velocity and that of smaller-scale wave drag with outflow velocity are calculated for the three cases of defining large-scale waves as 1–2, 1–3, and 1–4. Then the difference between the correlations for large-scale waves and for smaller-scale waves are calculated and profiles of these differences are shown in Fig. 4c, d. We expect the largest-scale waves to exhibit a stronger positive correlation within the deep branch region, while smaller-scale waves are anticipated to show a stronger positive correlation in the shallow branch region. Hence, the correlation difference profiles should be positive in the deep branch and negative in the shallow branch region, respectively. Including wave 4 into the large-scale waves results in a stronger correlation of the outflow velocity in the shallow branch region of the Southern Hemisphere with the large-scale waves compared to the smaller-scale waves (blue line in Fig. 4d). Additionally, incorporating wave 3 into the smaller-scale waves enhances the correlation between outflow velocity and smaller-scale wave drag at certain levels in the middle stratosphere, surpassing the correlation observed for the large-scale waves (red lines in Fig. 4c, d). Taking all these aspects into account, we conclude that the clearest separation between different circulation branches at upper and lower levels in terms of wave drag occurs between waves 3 and 4. Hence, we define the deep circulation branch of the stratospheric BDC to be driven by planetary waves 1–3 and the shallow circulation branch to be driven by smaller-scale waves (wavenumbers larger than 3). The following analyses of BDC structure, variability and trends, support our choice in separating the wave driving into these two classes of waves as separated by scale.

Vertical profiles of upwelling and outflow generated by planetary and smaller-scale waves were constructed for the different reanalyses and are presented in Fig. 5. Regardless of the reanalysis, planetary waves generate the majority of the outflow in the middle stratosphere and nearly all of the outflow in the upper stratosphere. Smaller-scale waves, on the other hand, are responsible for most of the outflow in the lower stratosphere. Compared to the other three reanalyses, ERA5 indicates a notably higher total outflow in the lower stratosphere between about 20-25km. This difference is likely related to the effects of small-scale waves and the outflow they generate. Specifically, better spatial and temporal resolution of ERA5 allows for better representation of mesoscale (gravity) waves in ERA5 than in the other reanalyses (eg. Hoffmann et al., 2019) . The drag caused by planetary waves is very similar between all four reanalyses. Hence, ERA5 is able to resolve more mesoscale waves and has a stronger related wave drag than other reanalyses. Consequently, in ERA5 these waves contribute significantly more strongly to the shallow branch.

Figure 5 further shows that the level which separates the two dynamical regimes with outflow mainly generated by planetary waves above and mainly by smaller-scale waves below emerges similarly for the different reanalyses. Therefore, this level represents a natural separation in terms of dynamical characteristics, i.e. wave driving, between the shallow and deep branches of the stratospheric circulation and is largely consistent across the different datasets considered.

### 3.3 Level of the separation between shallow and deep branches

The clear difference between the contributions of planetary and smaller-scale waves to the stratospheric residual circulation at upper and lower levels allows the robust definition of a separation level between a deep and shallow circulation branch in terms of wave drag. We define the separation level between the deep and shallow branches as the lowest altitude at which the contribution to outflow from planetary waves is equal to the contribution from smaller-scale waves. To eliminate sharp spikes of outflow in monthly mean data, which can cause unexpected variations in the separation level height, a rolling mean over five vertical levels is applied to the outflow profiles before determining the separation level. In cases where the monthly mean data shows equal contributions to the outflow from planetary and smaller-scale waves at multiple altitudes, we consider the separation level too vague for accurate estimation and exclude these months.

The seasonal variation of the separation level is shown in Fig. 6. The separation level shows a weak seasonal cycle with a somewhat lower separation level height in boreal summer months, similarly in all four reanalyses considered. The separation level height depends on the relative contributions of planetary waves and smaller-scale waves to tropical outflow. A stronger contribution from planetary waves raises the separation level, whereas a stronger contribution from smaller-scale waves lowers the separation level (Fig. 5 b). Therefore, the seasonal cycle of the separation level is likely linked to the annual cycle of wave drag. In particular the weaker planetary wave activity during boreal summer is likely important for pushing the separation level downward.

The large variability of the separation level (standard deviation, shown by the whiskers in Fig. 6) indicates a large inter-annual variation of the separation level height, which is of similar magnitude as the seasonal variation.

Furthermore, during most months the separation level is located at higher altitudes for ERA5 compared to other reanalyses (Fig. 6). This difference is likely related to the ability of ERA5 to better resolve smaller-scale waves which shifts the separation level to higher altitudes compared to the other reanalyses. Given its higher resolution and ability to resolve larger parts of the mesoscale wave spectrum, we consider the estimate of the separation level height from ERA5 likely more accurate compared to the other reanalyses.

The focus of this study is on the effects of resolved waves, but we notice that contributions from smaller-scale, unresolved waves could modify our results. Since smaller-scale waves are responsible for driving the shallow branch and since the upwelling deficit increases at lower levels in the shallow branch region (Fig. A2) it is likely that also the parameterized wave drag contributes significantly to the shallow branch. Therefore we expect the separation level to be located at somewhat higher altitudes for higher resolution data with a greater portion of the smaller-scale wave spectrum resolved or with parameterized wave drag included in the annalysis.

# 4 Variability and trends in circulation structure

## 4.1 Variability of the circulation

Figure 7 shows the upwelling at 100 hPa which is close to the tropopause, at 43 hPa which is approximately the separation level between shallow and deep branch, and the difference between upwelling at 100 hPa and 43 hPa. The upwelling at the tropopause represents the total mass flux into the stratosphere within the BDC and is related to the outflow at all levels above in the stratosphere (with a small portion of it even related to outflow at levels above the stratopause). Therefore, changes in upwelling at this level reflect changes in the overall overturning circulation. The annual mean upwelling at 100 hPa, generated by all resolved waves and estimated using the DWCP, is approximately $9.9 * 10^9$ kg/s in ERA5. In comparison, MERRA2 and ERA-Interim exhibit slightly weaker upwelling rates of about $9.5 * 10^9$ kg/s, while JRA55 shows the weakest upwelling at this level, of about $9.2 * 10^9$ kg/s. The upwelling variability at this level is substantial, with the standard deviations estimated from deseasonalized monthly mean upwelling ranging from $1.2 * 10^9$ kg/s to $1.4 * 10^9$ kg/s across the reanalyses (Tab. 1). At 100 hPa, the planetary waves and resolved smaller-scale waves contribute comparably to the annual mean upwelling, with approximately 53% attributed to planetary waves and 47% to smaller-scale waves. In contrast, the other reanalyses exhibit a slightly greater contribution from planetary waves ranging from 58% for ERA-Interim to 61% for MERRA2. The upwelling at 100 hPa caused by planetary wave drag has a standard deviation comparable to that caused by smaller-scale waves (Tab. 1). Therefore, both planetary and smaller-scale waves contribute comparably to the forcing and variability of the whole overturning circulation.

Lin et al. (2013) divided the stratospheric circulation into a deep branch (above 30 hPa), a shallow branch (between 70 hPa and 30 hPa) and a transitional branch (between 100 hPa and 70 hPa). Although the lowest layer (100-70 hPa) is still influenced by tropospheric processes, our results show no clear indication of differences in wave driving for this layer compared to the layers above. Therefore, in the present study we aim at a dynamical separation of only the shallow and deep branches based on the circulation's wave driving.

The total outflow of the shallow branch can be estimated by subtracting the upwelling at the separation level (43 hPa) from the upwelling at the bottom of the stratosphere (100 hPa), as shown in Fig. 7b. In this 100-43 hPa layer, dissipation of the resolved waves generates an annual mean total outflow ranging from approximately $5.3 * 10^9$ kg/s to $5.8 * 10^9$ kg/s depending on the reanalysis. Roughly two-thirds of this outflow can be attributed to smaller-scale waves, while only about one-third is attributed to planetary waves. However, the monthly mean standard deviations of the total outflow driven by planetary waves and smaller-scale waves indicate that both waves types contribute comparably to the variability of the total outflow in this layer, and hence to the variability of the shallow circulation branch.

The upwelling across the mean separation level is a result of the wave drag in the deep branch region above that level. The annual mean of this upwelling across the separation level is ranging from $3.9 * 10^9$ kg/s to $4.1 * 10^9$ kg/s depending on the reanalysis, the majority of which is caused by planetary waves and less than 20% by smaller-scale waves. The variability of the monthly mean upwelling across the separation level (43 hPa) is mostly controlled by planetary waves (see Fig. 7). This is indicated by the fact that variability (here measured in terms of standard deviation) of upwelling induced by planetary waves is more than two times larger compared to the variability of upwelling induced by smaller-scale waves (Tab. 1). Therefore, despite

**Table 1.** Climatological mean upwelling and standard deviation of deseasonalized monthly upwelling in $10^9$ kg/s. All values are based on ERA5 1980-2017 data.

| level | total | | planetary | | smaller-scale | |
|---|---|---|---|---|---|---|
| | $W$ | $\sigma$ | $W$ | $\sigma$ | $W$ | $\sigma$ |
| ERA5 | | | | | | |
| 43 hPa | 4.09 | 0.88 | 3.35 | 0.80 | 0.73 | 0.33 |
| 100 hPa - 43 hPa | 5.79 | 1.14 | 1.91 | 0.97 | 3.88 | 0.82 |
| 100 hPa | 9.88 | 1.24 | 5.27 | 1.07 | 4.61 | 0.80 |
| ERA-Interim | | | | | | |
| 43 hPa | 3.87 | 0.88 | 3.22 | 0.78 | 0.64 | 0.32 |
| 100 hPa - 43 hPa | 5.58 | 1.17 | 2.23 | 0.97 | 3.35 | 0.79 |
| 100 hPa | 9.45 | 1.27 | 5.45 | 1.1 | 4.00 | 0.76 |
| JRA55 | | | | | | |
| 43 hPa | 3.92 | 0.92 | 3.33 | 0.83 | 0.60 | 0.31 |
| 100 hPa - 43 hPa | 5.29 | 1.16 | 2.04 | 0.98 | 3.24 | 0.79 |
| 100 hPa | 9.21 | 1.23 | 5.37 | 1.06 | 3.84 | 0.74 |
| MERRA2 | | | | | | |
| 43 hPa | 4.05 | 0.90 | 3.43 | 0.82 | 0.62 | 0.33 |
| 100 hPa - 43 hPa | 5.50 | 1.36 | 2.41 | 1.21 | 3.09 | 0.83 |
| 100 hPa | 9.55 | 1.42 | 5.84 | 1.23 | 3.71 | 0.83 |

the clear separation in wave driving concerning the climatological mean circulation, both wave types contribute significantly to the circulation variability of the shallow branch, while the variability of the deep branch is mostly controlled by the planetary waves.

To further investigate the roles of planetary and smaller-scale waves in driving circulation variability, vertical profiles of standard deviation of tropical outflow attributed to planetary and smaller-scale waves were constructed, as well as vertical profiles of the correlation coefficients between the total outflow and the contributions by planetary or smaller-scale waves (Fig.

8). Planetary waves are relevant to the variability of the circulation throughout the stratosphere and generally contribute more to the overall variability of the circulation compared to smaller-scale waves. In the middle and upper stratosphere these large-scale waves clearly dominate the circulation variability, consistent with a particularly high standard deviation of upwelling (Tab. 1) At levels below about 25km, the contribution of planetary waves to the variability of the circulation weakens, while the contribution of smaller-scale waves increases, such that both wave types contribute comparably (Fig. 8). The generally

higher impact of planetary waves on the variability of the circulation is possibly related to the fact that, the standard deviation of the upwelling at 100 hPa takes into account changes in the wave drag from that level and all levels above (see Sect. 2).

Filtering by the background wind according to the Charney-Drazin criterion (Eq. 9) more strongly affects the smaller-scale waves, hinders their upward propagation and therefore dampens their contribution to circulation forcing at upper levels. Hence, the wave drag of the planetary waves is distributed over a larger altitude range in the stratosphere compared to the wave drag of the smaller-scale waves. In summary, planetary waves significantly contribute to the variability of the circulation at all levels while smaller-scale waves only affect the circulation variability in the shallow branch region.

## 4.2 Trends

Multi-decadal trends in the upwelling over the period 1980–2017 are presented in Fig. 9. Given the relatively short time period for trend calculation, upwelling trends are mostly not statistically significant at 95% confidence level. It is known that BDC trends before and after about the year 2000 are affected by the opposite effects of ozone depletion and ozone recovery (e.g. Polvani et al., 2018; Abalos et al., 2019; Fu et al., 2019). Ozone depletion contributes to a BDC strengthening before 2000, strongest in the SH, while ozone recovery contributes to BDC weakening. However, dividing the analysis into the two sub-periods 1980-2000 and 2000-2017 did not enhance the significance of trends (not shown).

The trends of upwelling estimated from the TEM residual circulation velocity (black dotted lines in Fig. 9) and from the DWCP (black solid lines in Fig. 9) can differ significantly at different levels. While here, we mostly focus on the effects of resolved wave drag, most previous studies have concentrated on total upwelling derived from the TEM velocities (e.g. Fujiwara et al., 2022). Although the upwelling trends differ strongly between different altitude ranges, several consistent patterns in the trend profiles emerge for the different reanalyses. Above 25 km, the upwelling trend estimated from the total wave drag is positive and S-shaped for all four reanalyses with a local trend maximum around 35km. Strongest differences to the other reanalyses are found for MERRA2. In particular, at lower levels the shallow branch trend is mostly weakly positive for MERRA2, but negative for the other three reanalyses. For ERA5, ERA-Interim and JRA55 the deep branch trend is largely driven by planetary waves while for MERRA2 smaller-scale waves also contribute comparably to the trend. Furthermore, MERRA2 shows a generally more positive trend of upwelling driven by smaller-scale waves than that driven by planetary waves below about 30km, which is opposite to the other three reanalyses. For most reanalyses and levels the agreement between trends of upwelling estimated from the total resolved wave drag and from the TEM approach is poor. Only for ERA5 the trend from resolved wave drag matches the total TEM velocity-based trend well between about 23-35km. Also, ERA5 is the only reanalysis where the trend in the shallow branch is almost exclusively controlled by the smaller-scale waves, and which shows a significant contribution of resolved mesoscale waves to the overall trend of the shallow branch.

The robust difference in the contributions of planetary and smaller-scale waves to dominate the upwelling trends above and below about 22km (Fig. 9) corroborates our wave drag-based estimate of the separation level between the shallow and deep branch layers. In particular, different trends in dynamics are apparent in different altitude regions, indicating a decoupling of the two circulation branches regarding their long-term evolution. These differences in trends between the shallow and deep branch are consistent, at least qualitatively, for the different reanalyses.

As noted before, the upwelling at a given level is driven by the wave drag at the level and all levels above it and hence the upwelling trend at a given level is not related unambiguously to wave drag changes only at that level. A clearer relation to wave

drag changes at a given level holds for changes in meridional outflow at that level. Trend profiles for outflow are presented in Fig. 10 for the four different reanalyses and separated into contributions from smaller-scale and planetary waves. Given the relation to upwelling in Eq. 8, trends in outflow are directly related to the vertical derivative of the trends in upwelling. Therefore, the outflow trends are positive in a shallow layer above the tropical tropopause. Above, outflow shows generally negative trends throughout a broad layer between about 18-35km (somewhat depending on the reanalysis considered), and indicating weakening of transport out of the tropics in that altitude region. Above about 35km, outflow increases throughout an about 5km thick layer, consistent with the negative vertical gradient in upwelling trends in Fig. 9. Whereas most of the reanalysis-based trends are statistically insignificant, we hypothesize that the robust vertical pattern, with outflow weakening below and strengthening above about 35km, could be indicative for an upward shift of the deep branch of the stratospheric circulation. Another robust vertical pattern is the strengthening of outflow in the lower part of the shallow branch and weakening above, which could be indicative for a downward shift of the shallow branch of the stratospheric circulation. Overall, longer time-series are needed to deduce significant long-term trends of the stratospheric circulation from reanalysis data.

Given the limited period of the analyzed data and large inter-annual variability of the separation level height, the trend of the separation level for any of the reanalyses is smaller than two standard errors and therefore not significantly different from zero. However, three out of four reanalyses show negative trends of the separation level height, while only JRA55 has a slightly positive trend (Fig. 11). The negative trend of the separation level can also be estimated from Fig. 10, which shows that the trend of large scale waves is generally more positive than the trend of small scale waves around the separation level. Hence the contribution of planetary waves around the separation level increases over time compared to the contribution of smaller-scale waves, and consequently pushes the separation level slightly downwards.

## 5 Discussion

Past research has shown that the stratospheric circulation can be separated into a deep and a shallow branch (e.g. Birner and Bönisch, 2011; Plumb, 2002). However, different metrics have been used in past studies to separate these circulation branches. Some studies applied a separation by selecting a boundary pressure level which splits the upwelling mass flux approximately into a specific ratio between the deep and shallow branch, e.g. ∼50 hPa (Ball et al., 2016) or 30 hPa (Lin et al., 2013; Diallo et al., 2019). Since vertical velocities in the stratosphere are much weaker than horizontal ones, transit times from the stratospheric entry point at the tropical tropopause to the downwelling region in the extratropical lowermost stratosphere along the shallow branch are significantly lower than along the deep branch. Based on this difference in transit time, a separation between deep and shallow circulation branches has also been proposed based on transit times along residual circulation trajectories (Birner and Bönisch, 2011). Furthermore, the transport of air masses from the so-called "tropically controlled transition region" with an upper boundary at about 450K (approximately 70 hPa) into the extratropics is likely related to the shallow branch (Rosenlof et al., 1997; Bönisch et al., 2009; Bönisch et al., 2011). In view of such diversity in proposed separation between the shallow and the deep stratospheric circulation branch, here we aim for a dynamical separation

based on differences in the circulation's wave driving as inferred from reanalysis data. The robustness of our results is estimated from comparison of four different reanalyses.

It is well established that the deep branch of the BDC is primarily driven by planetary waves which break in the subtropical middle and upper stratosphere, while the shallow branch is more strongly driven by synoptic waves which break in the subtropical lower stratosphere (e.g. Plumb, 2002). Our statistical and mechanistic analyses show that in current global reanalyses the outflow in the deep branch is indeed largely driven by planetary waves with wavenumbers 1-3. While the shallow branch, is mainly driven by synoptic scale waves (wavenumbers 4 to 20) and to a lesser extent by mesoscale waves (wavenumber larger

than 20). It should be noted, that due to coarse resolution in the current generation reanalyses the contribution of mesoscale, mainly gravity, waves in our study is likely underestimated, consistent with a significant DWCP upwelling deficit (Fig. A2 and also with results by Butchart et al. (2011)). For the resolved waves considered here, however, the wave drag-based separation of the deep and the shallow branch is robust across all four reanalyses used.

    Based on the present study the mean level separating the deep and the shallow branch of the residual circulation is located

between 43 and 47 hPa depending on the reanalysis. Hence, despite using different arguments the separation level found in this work is close to the levels estimated in past studies. Moreover, this separation level between the two circulation branches was found to have an annual cycle, with lowest separation level during boreal summer, which is similar in all reanalyses used in this work. Both the inter-annual and seasonal variability of the separation level have an amplitude which is larger than the differences between the reanalyses.

Most of the residual circulation trends estimated in this work are within two standard errors around zero and therefore are not statistically significant. Yet there are patterns which robustly emerge in all reanalyses used, as an acceleration of upwelling above 28 hPa, and mostly negative trends of outflow between 60 and 15 hPa. Remarkably, also the planetary wave contribution to the trends varies considerably between reanalyses, indicating that even large-scale waves are not sufficiently constrained in reanalyses with respect to trends. It should be noted that this study focuses solely on the resolved wave drag. Depending on the

reanalysis, inclusion of the parameterized wave drag could significantly influence the results by increasing the contribution of smaller-scale waves to the circulation, which would likely shift the separation level upward.

    The reanalyses data used here were down-sampled to 1° x 1° horizontal resolution and 6-hour temporal resolution. Figure 12 presents upwelling and outflow estimated using both, fine resolution (0.3° x 0.3° x 1 hr) and coarse resolution (1° x 1°, 6 hr) ERA5 data for the year 2010. The overall structure of the vertical profiles of upwelling remains largely unaffected by the

490 down-sampling of the reanalysis data. The most notable effect of down-sampling is found for waves with wavenumbers greater than 180 for which all information is lost. For the original, high-resolution data, these very small-scale waves add a significant contribution to the driving of the shallow BDC branch in the lower stratosphere. Upwelling near the tropopause associated with waves 21 to 180 is reduced by about 20%. Consequently, the total upwelling at that level decreased by about 1.5%. Similarly, outflow near the separation level generated by waves 21 to 180 is reduced by about 5%, with a corresponding total outflow

reduction of around 1.3%. The separation level height has decreased by about 0.2 km or 1. hPa.

    Overall, the down-sampling to coarser spatial resolution did not significantly affect the large-scale features of the residual circulation considered here (see Fig. A4 and Fig. A5). As also shown by Seviour et al. (2011), from a climatological monthly-

mean perspective the 6-hourly data is sufficient to take tidal effects into account. Furthermore, for 6-hourly temporal resolution some distortions remain in monthly mean data for $\bar{w}^*$ (Fig. A5), while only little to no distortions remain for $\nabla \cdot F$ (Fig. A4).

Regarding daily mean data for specific dates, reducing the temporal resolution from 1 to 6 hours has a significant impact on the representation of the residual circulation. Therefore, in order to investigate short-term and smaller-scale features of the circulation it seems important to use higher temporal resolution than 6 hours. Nevertheless, the overall characteristics of the circulation on the monthly timescale are affected by the down-sampling only to a limited and acceptable extent.

Vertical profiles of upwelling estimated using the TEM approach for all four reanalyses are shown in Fig. A2 together with

the difference to the DWCP upwelling estimate ($(W - W_D)/W * 100\%$ denoted relative "upwelling deficit" in the following), which is used to estimate the contribution of parameterized wave drag to the upwelling. For all four reanalyses, the vertical profile of the upwelling deficit shows a C-shape, with a higher deficit both in the lowest and upper parts of the stratosphere. This C-shape of the deficit indicates that, the parameterized wave drag in the reanalyses mostly affects lowest and highest parts of the stratosphere. This effect is likely related to the contribution of unresolved gravity wave drag to both the mesospheric

circulation and to the shallow branch of stratospheric circulation (e.g. Diallo et al., 2019; Eichinger et al., 2020), which is not fully resolved in the reanalysis data. Due to its higher resolution and greater ability to resolve mesoscale gravity waves, ERA5 shows significantly stronger mesoscale wave drag (cf. Fig. 5 and Sect. 3). Thus, ERA5 has the smallest upwelling deficit in the lower and upper stratosphere. In the middle stratosphere the difference between ERA5, MERRA2 and ERA-Interim is small, likely related to the small contribution of mesoscale waves at these levels. Throughout the stratosphere, JRA55 shows

the largest deficit among all four reanalyses considered.

Furthermore, ERA5 shows the strongest total resolved upwelling and outflow at the bottom of the stratosphere (see Fig. A3b). This indicates that the total resolved wave drag is about 5 to 10% greater in ERA5 than in the other reanalyses. The total TEM-based upwelling at the same altitude, including effects from parameterized waves, is significantly weaker in ERA5 than in ERA-Interim or JRA55, therefore results are consistent with Ploeger et al. (2021). At the top of the stratosphere, the TEM

upwelling in ERA5 is 20 to 30% weaker than in the other reanalyses (see Fig. A3a).

In this paper we mainly focus on the circulation driven by the resolved wave drag. In general, the upwelling based on the different reanalyses is more similar when estimated from resolved waves only (Fig. A3a, b). When considering the upwelling deficit, the difference between the full TEM–based and the DWCP–based upwelling, differences between the reanalyses may indicate differences in the gravity wave parameterization. In particular, ERA5 has generally stronger upwelling estimated from

the resolved wave drag and weaker TEM upwelling in the regions where the effect of small-scale gravity wave drag on the circulation is expected to be strongest. Therefore, the smaller upwelling deficit in ERA5 is likely due to a better ability to resolve the waves and also from weaker parameterized wave drag.

Finally, it should be noted that throughout this paper we have used a scale–based separation of atmospheric waves. An additional important question would be a separation of the waves into physical wave classes like Rossby and gravity waves.

Although the scale separation does not necessarily represent a physical classification, it provides some insights into the physical types of waves. Planetary waves 1–3, as found here to constitute the driving force of the deep circulation branch, are large-scale Rossby waves. Comparison of $\nabla \cdot F$ with studies based on physical wave models (eg. Koval et al., 2023) show that these large-

scale Rossby waves in the lower to middle stratosphere are mostly stationary or quasi-stationary. Furthermore, the mesoscale waves with wavenumbers 20-180, as found here to drive mainly the shallow branch, are likely associated with gravity waves. Since in the lower stratosphere Kelvin waves are confined to the tropics and typically do not extend to the vicinity of the TAL where the main driving of tropical upwelling occurs, they are likely of limited relevance for the present study (Ern and Preusse, 2009; Yang et al., 2003; Smith et al., 2002). At higher altitudes above the stratopause, a broader range of physical wave types becomes increasingly important (cf. Koval et al., 2023; Yasui et al., 2021). Since reanalyses data do not inherently distinguish between wave modes, a precise quantification of contributions from different physical wave types is beyond the scope of this study. However, incorporating a more physically based separation of wave types would be a valuable direction for future research.

## 6 Conclusions

Based on both statistical and a mechanistic analysis we here propose a dynamical separation of the deep and shallow branches of the stratospheric circulation. The deep branch of the BDC is mainly driven by large-scale planetary waves with wavenumbers 1 to 3, while the shallow branch is driven by smaller-scale waves with wavenumber greater than 3. The mean altitude of the level separating the deep and shallow branch is at about 22 km (43 hPa) with significant inter-annual and seasonal variations. In particular, the separation level height shows a weak semi-annual cycle in all four reanalyses used in this work. The dynamical separation criterion found in this work is observed to be robust across all four reanalyses.

At the tropical tropopause, both wave types (planetary and smaller-scale) contribute comparably to the upwelling and both branches of the overturning circulation move a similar amount of mass. ERA5 reanalysis shows the strongest contribution from smaller-scale waves to the driving of the BDC while for the other three reanalyses the contribution from smaller-scale waves is somewhat weaker. While the variability of the deep branch is mostly controlled by planetary waves, both planetary and smaller-scale waves affect the variability of the shallow branch to a similar extent.

The circulation trends over the period 1980-2017 estimated here for the different reanalyses were found to be statistically insignificant. Furthermore, dividing the period into ozone recovery and post-ozone recovery phases did not notably enhance the statistical significance of the trends. However, some of the trend features similarly emerged for all four reanalyses considered. The robust features concern a strengthening of the circulation's tropical outflow at altitudes of about 35-40 km and a weakening below, potentially indicating an upward shift of the deep circulation branch. At lower levels just above the tropical tropopause, a strengthening of tropical outflow related to the shallow branch emerged robustly for the different reanalyses.

Given the differences between reanalyses regarding the climatological height, variability and trends in the separation level, for detailed investigations of the structure of the stratospheric circulation a precise identification of the separation level is needed. In particular, calculating the specific separation level for a given reanalysis or model dataset, such as by using the wave drag approach proposed in this study, appears to be advantageous for model inter-comparisons. Otherwise, we recommend using the separation level calculated here from ERA5 (Tab. **??**), given the ability of that reanalysis to better represent effects of small-scale waves compared the older reanalyses.

It should also be noted that the 70 hPa level, which is often used as a reference level for model inter-comparisons of stratospheric tropical upwelling, is not unambiguously related to a distinct circulation branch but is indeed affected by both the deep and the shallow branch. As shown here, the separation level between these circulation branches may significantly differ between different models, and so may the effects of the deep and shallow branch on upwelling across 70 hPa. Using the appropriate separation levels for distinguishing the deep and shallow branch in model and reanalysis inter-comparisons could reduce the spread between models regarding climatology and trends in the stratospheric circulation.

. *Data availability.* The ERA5, and ERA-Interim, reanalyses data are processed by the European Centre for Medium-Range Weather Forecasts. ERA5 data is available from the Climate Data Store (https://cds.climate.copernicus.eu/). The JRA55 reanalysis data are processed by the Japan Meteorological Agency and available from the Data Integration and Analysis System Program (https://diasjp.net/en/). The MERRA2 data are processed by the NASA Goddard Global Modeling and Assimilation Office and available from the Data and Information Services Center (https://disc.gsfc.nasa.gov/).

. *Author contributions.* RB carried data processing and analysis. FP contributed code for the analysis. FP, PP, ME, and TB provided helpful discussions and contributed to the design of the analysis.

. *Competing interests.* The authors declare that they have no conflict of interest.

. *Financial support.* The article processing charges for this open-access publication were covered by the Forschungszentrum Jülich.

. *Acknowledgements.* FP acknowledges support by the Deutsche Forschungsgemeinschaft (DFG, German Research Foundation) – TRR 301 – project ID 428312742. Finally, we gratefully acknowledge the computing time for the CLaMS simulations which was granted on the supercomputer JUWELS at the Jülich Supercomputing Centre (JSC) under the VSR project ID CLAMS-ESM.

## Appendix A

This appendix contains auxiliary figures that provide more information about the contribution of individual zonal wavenumbers (Fig. A1), upwelling deficit and comparison between different reanalyses with ERA5 as a reference (Figs. A2, A3), and a comparison between full resolution ERA5 data and ERA5 data downsampled to resolutions of $1°$ spatially and 6 hours in time (Figs. A4, A5).

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

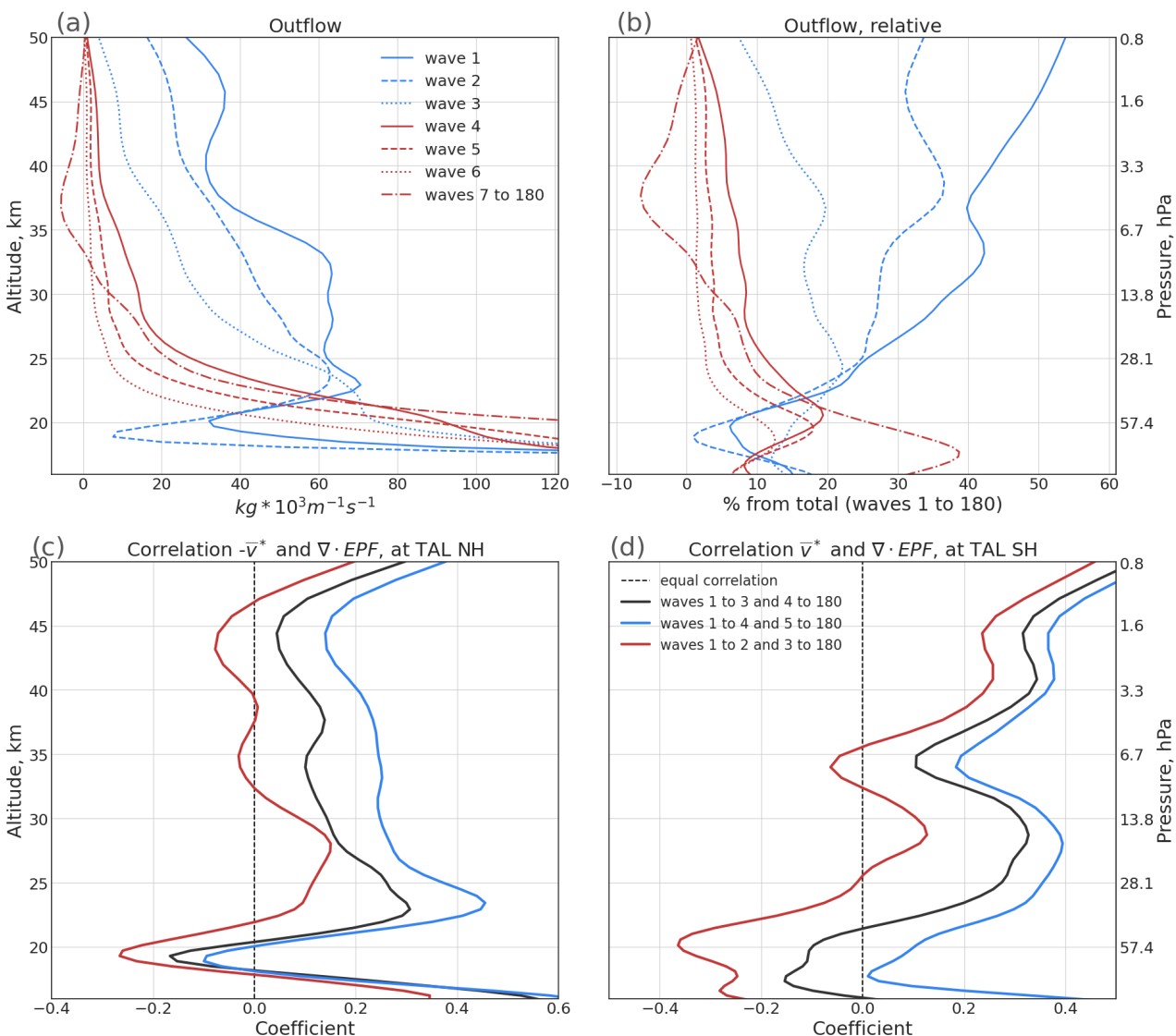

**Figure 4.** (a) Tropical outflow from the area between TAL into the extratropics calculated from Downward Control principle for different waves (colors and line styles). (b) Equivalent to (a) but in relative units as percentage of the total outflow of all resolved waves. (c) Difference between the correlations of outflow velocity from the TEM approach with wave drag generated by large-scale and smaller-scale waves (computed as correlation with large-scale minus correlation with smaller-scale), evaluated at turn-around latitudes in the Northern Hemisphere. Colors represent different wave bands for the large- and smaller-scale wave categories. (d) Same as (c) but for the southern hemisphere. Deseasonalized monthly mean data is used for calculation of the correlations presented in panels (c) and (d). The lines presented in this figure were smoothed with a Gaussian filter for better readability. The figure is constructed from monthly mean ERA5 data for 1980-2017.

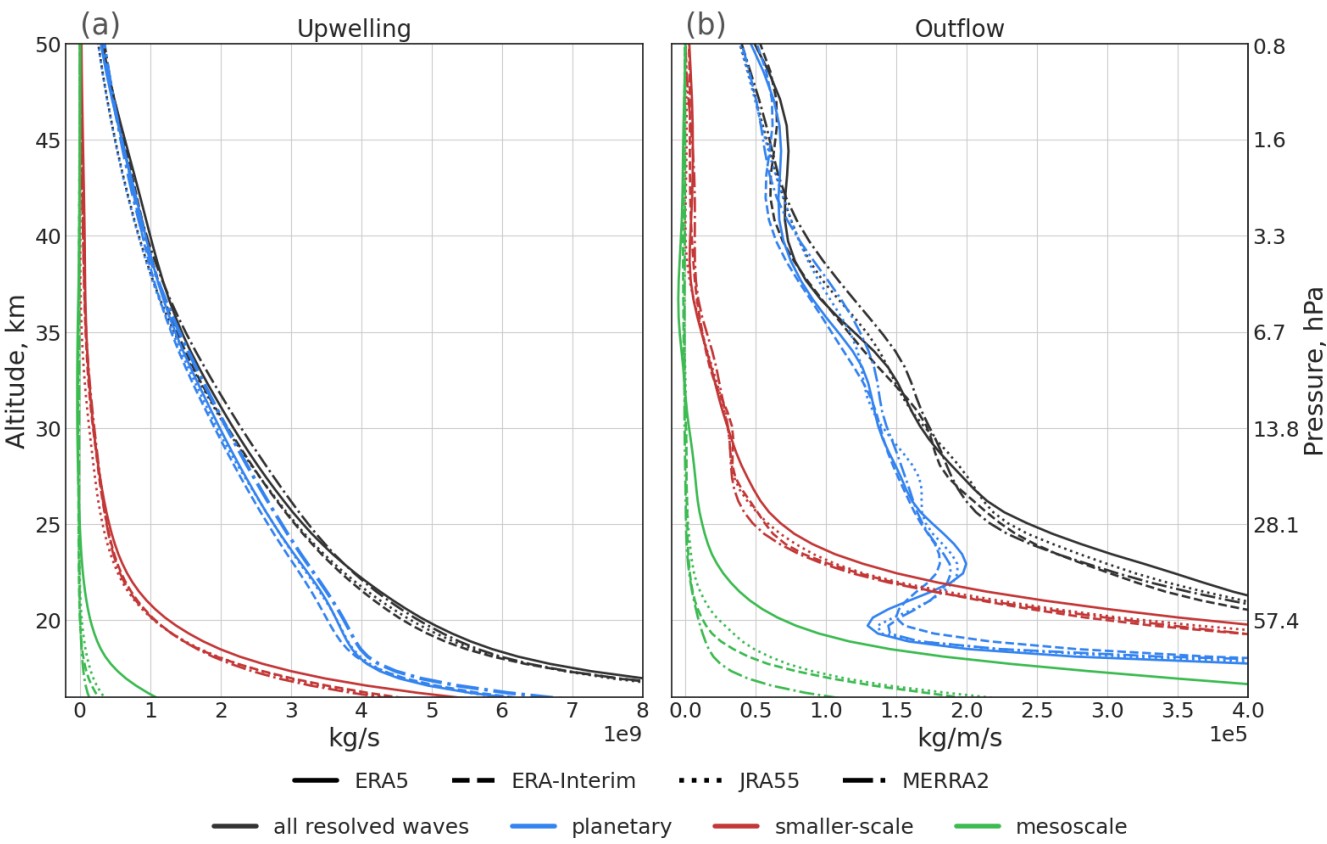

**Figure 5.** (a) Mean profiles of upwelling calculated from DWCP, induced by all resolved waves (black lines), planetary waves (blue lines), smaller-scale waves including mesoscale (red lines), and mesoscale waves (green lines) from ERA5 (solid), ERA-Interim (dashed), JRA55 (dotted), and MERRA2 (dash-dotted) reanalyses for 1980-2017. (b) Corresponding profiles of outflow. The lines presented in this figure were smoothed with a Gaussian filter for better readability.

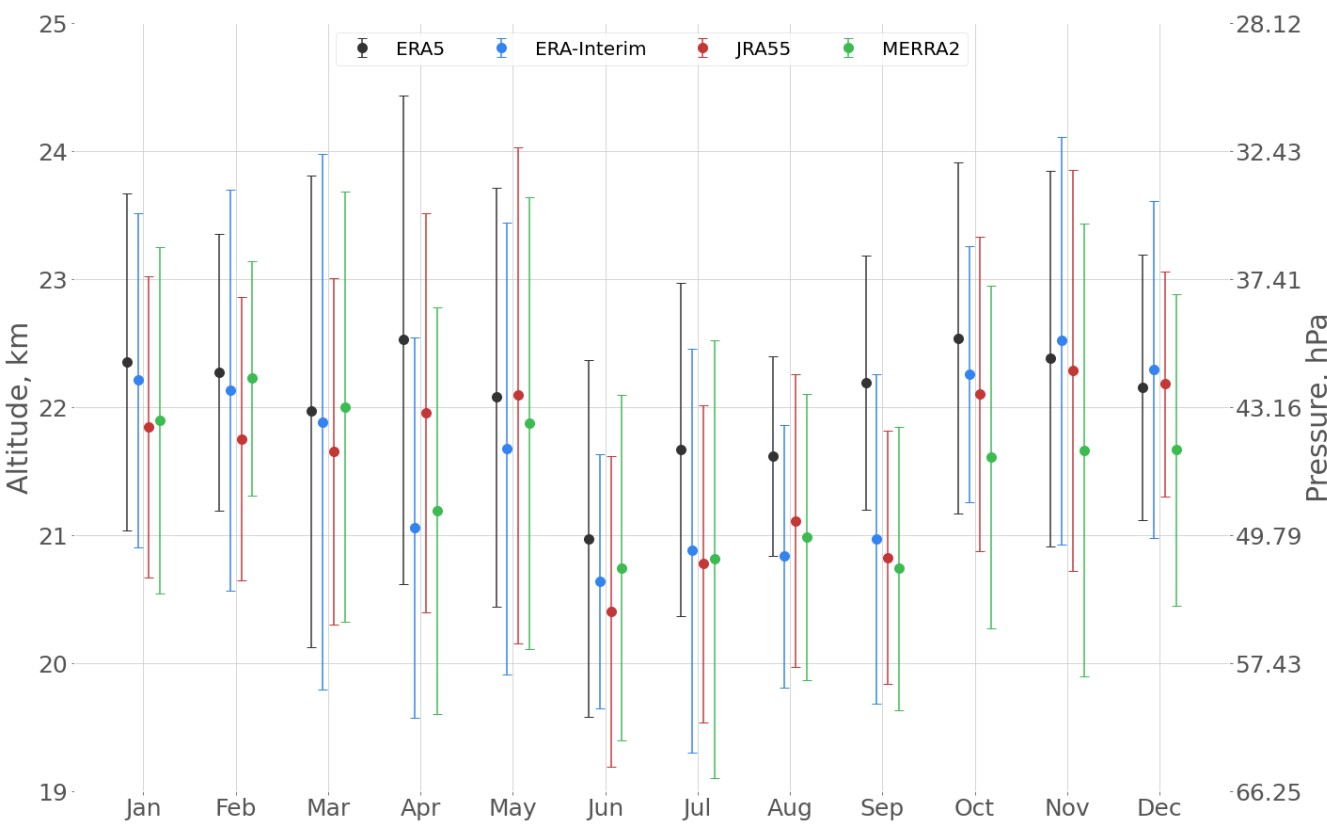

**Figure 6.** Seasonal cycle of the level separating the deep and shallow branches of the stratospheric circulation. Dots show the mean altitude of the level for each month, with the corresponding standard deviation as error bar. The colours represent different reanalyses: ERA5 (black), Era-Interim (blue), JRA55 (red) and MERRA2 (green). The level is estimated from 1980-2017 monthly mean data.

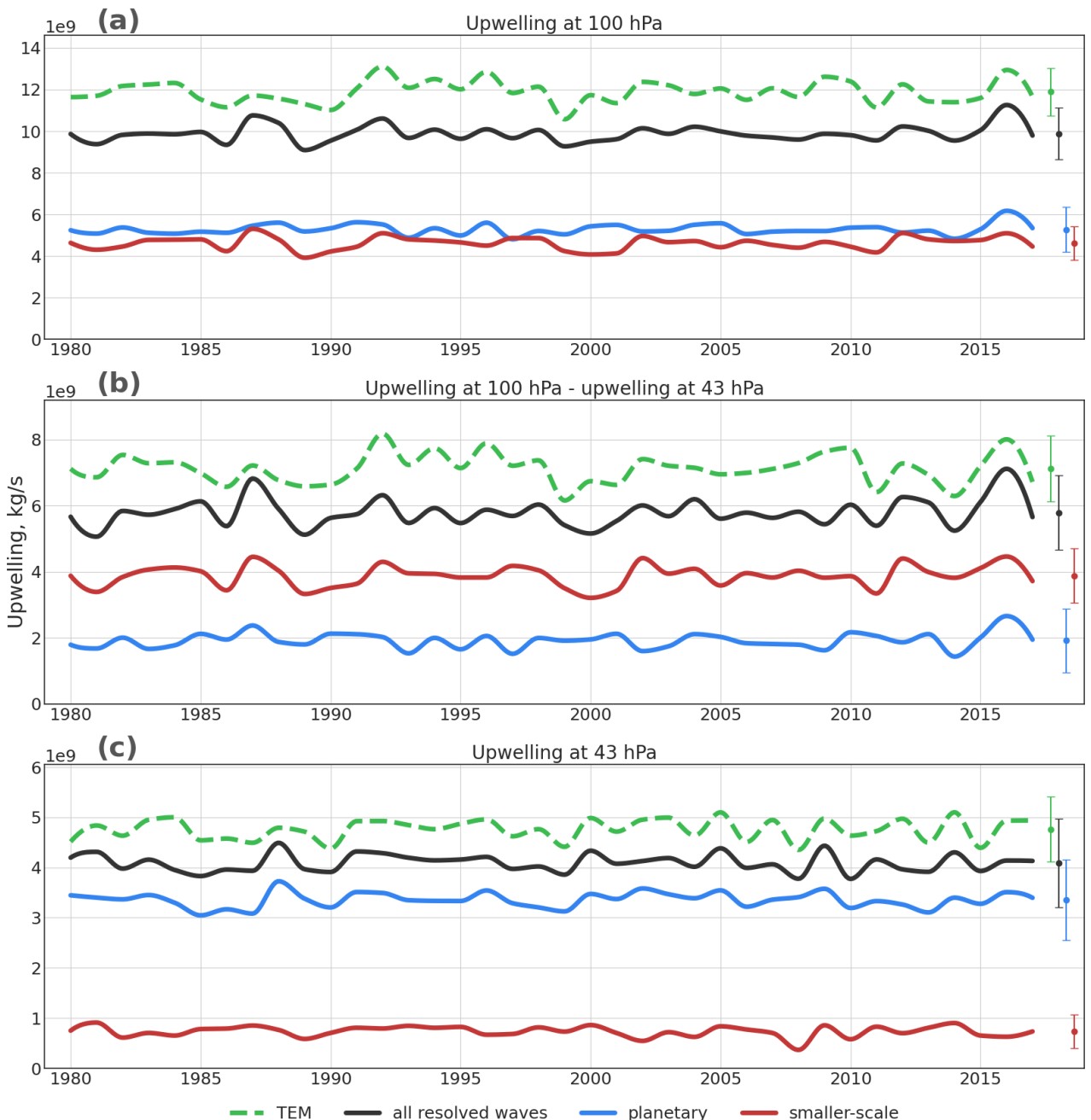

**Figure 7.** (a) Annual mean time series (1980–2017) of upwelling at 100 hPa based on ERA5 data. Shown are estimates from the TEM approach (green), total resolved wave drag (black), planetary wave drag (blue), and smaller-scale wave drag (red). Dots indicate mean values over the entire period, and error bars represent the standard deviation of monthly mean values, consistent with Tab. 1. (b) Difference in upwelling between 70 hPa and 43 hPa, using the same color scheme as in panel (a). (c) Upwelling at 43 hPa, again using the same legend as in panel (a).

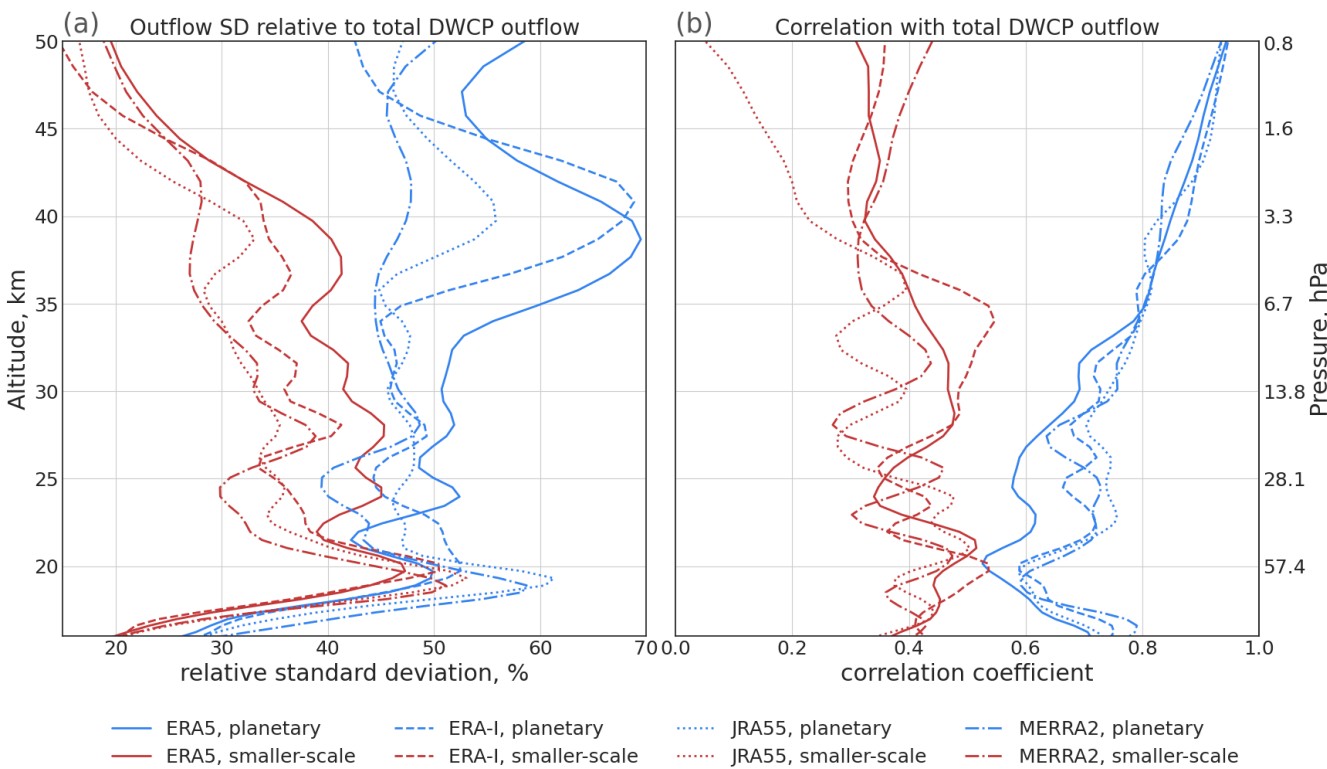

**Figure 8.** (a) Vertical profiles of tropical outflow variability, quantified by the standard deviation of DWCP outflow (blue lines driven by planetary wave drag, red lines by smaller-scale wave drag) relative to DWCP outflow driven by total resolved wave drag. (b) Vertical profiles of correlation coefficients between DWCP outflow driven by all waves and the outflow driven by only planetary (blue) and smaller-scale waves (red). Profiles have been constructed from 1980-2017 deseasonalized monthly mean data. The lines were smoothed with a Gaussian filter for better readability.

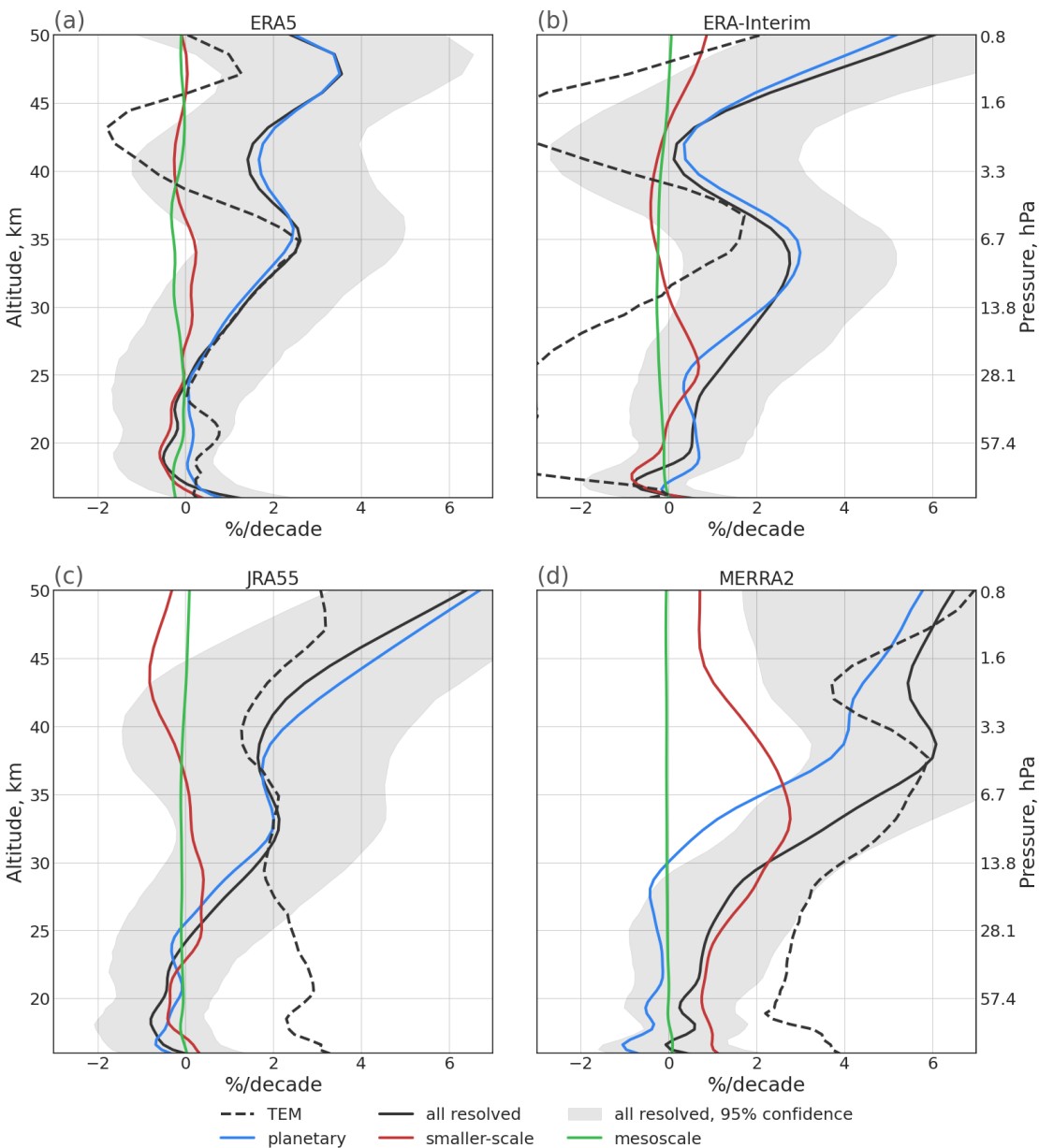

**Figure 9.** Vertical profiles of upwelling trend derived using TEM (dotted black line) or DWCP from total (black), planetary (blue), smaller-scale (red), and mesoscale (green) waves, gray shading denotes 95 percent confidence interval for total waves. The profiles were constructed from annual mean data for ERA5 (a), ERA-Interim (b), JRA55 (c), and MERRA2 (d). The lines were smoothed with a Gaussian filter.

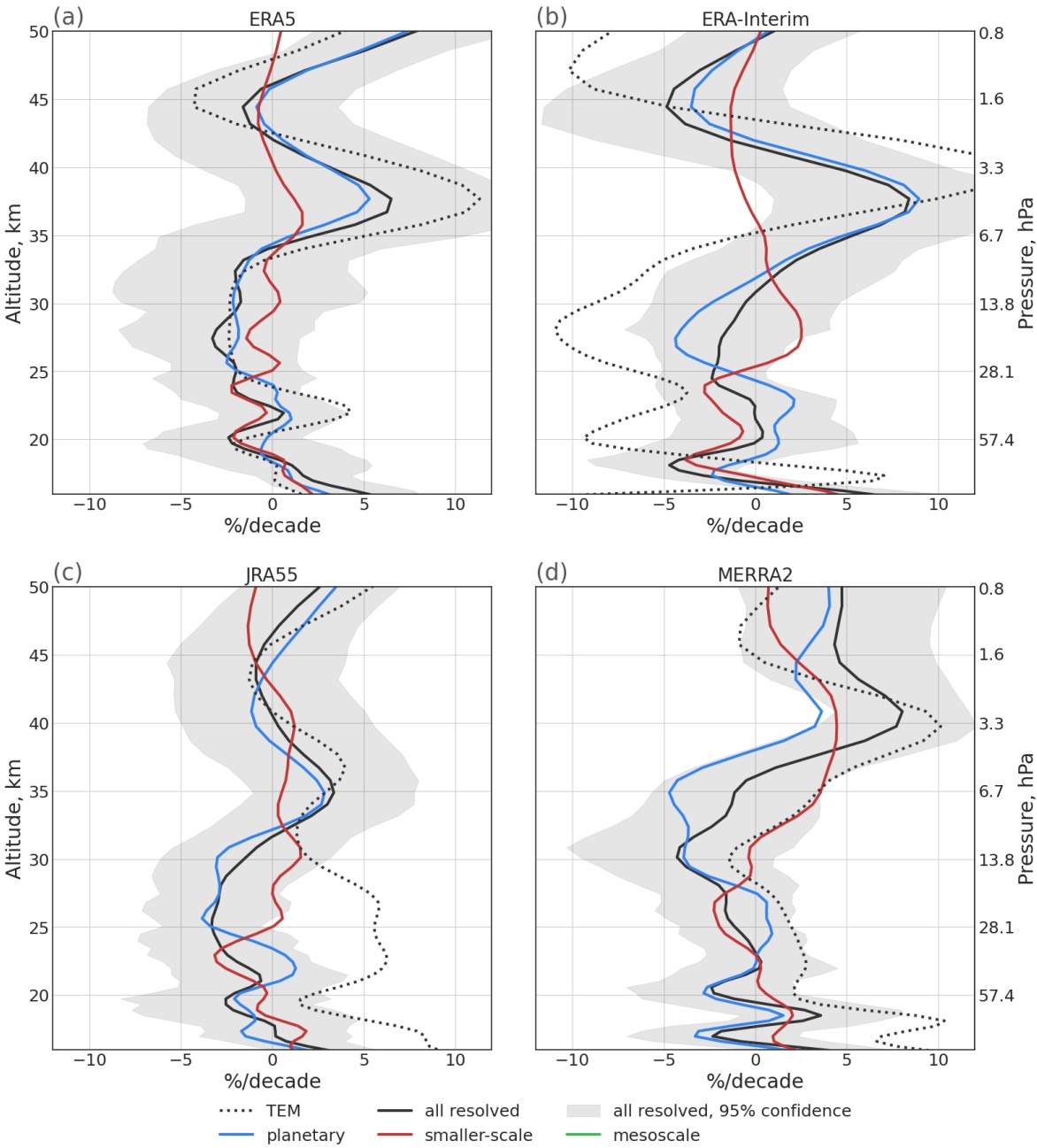

**Figure 10.** Vertical profiles of tropical outflow trend derived using TEM (dotted black line) or DC from total (black), planetary (blue), and smaller-scale (red) waves, gray shading denotes 95 percent confidence interval for total waves. The profiles were constructed from annual mean data of ERA5 (a), ERA-Interim (b), JRA55 (c), and MERRA2 (d). The lines were smoothed with a Gaussian filter.

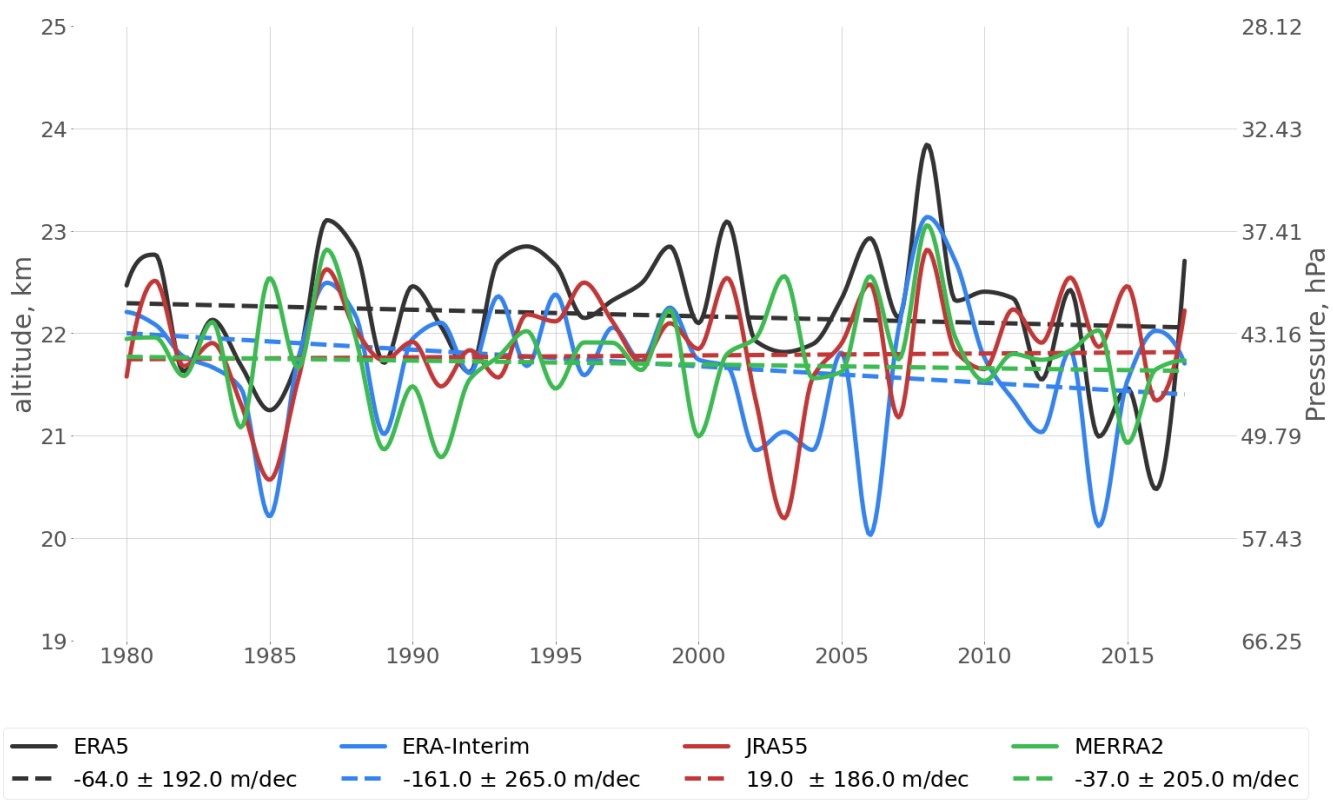

**Figure 11.** The variability and trend of the separation level between shallow and deep stratospheric circulation branches over the period 1980-2017 from ERA5 (black), ERA-Interim (blue), JRA55 (red), and MERRA2 (green). Solid lines represent the annual mean separation level and dashed lines show the linear trend based on the annual mean level. Decadal trend values ±2 standard error are given in the legend.

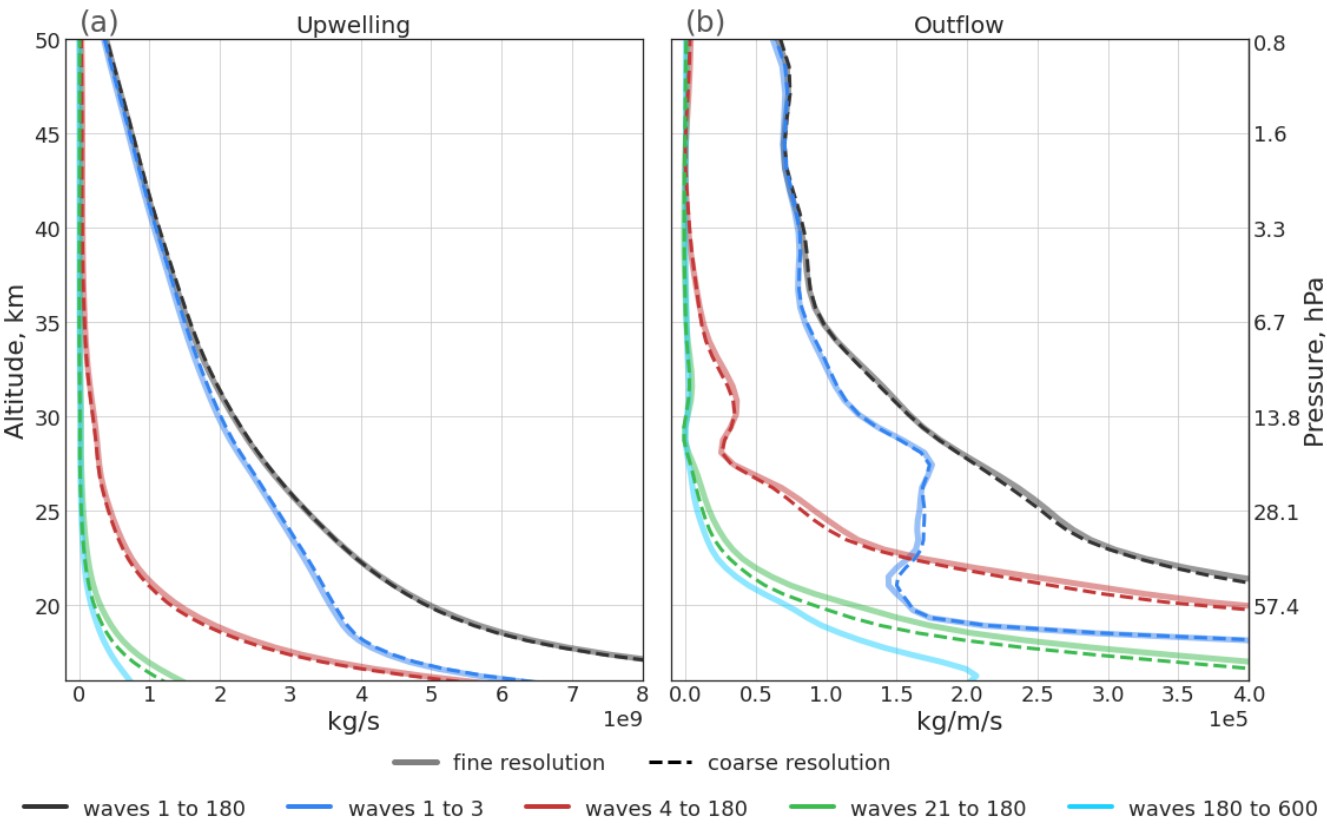

**Figure 12.** (a) Vertical profiles of mean upwelling estimated with DWCP from fine (solid lines, 0.3°, 1hr) and coarse (dashed lines, 1°, 6hr) resolution ERA 5 data for year 2010 and driven by all resolved wave drag (black lines), planetary wave drag (blue lines), smaller-scale wave drag (red lines), mesoscale wave drag (green lines), and small-scale mesoscale wave drag (cyan line). (b) equivalent to (a) with vertical profiles of mean outflow. As in the other figures, the TAL are estimated from coarse resolution data. The lines presented in this figure were smoothed with Gaussian filter.

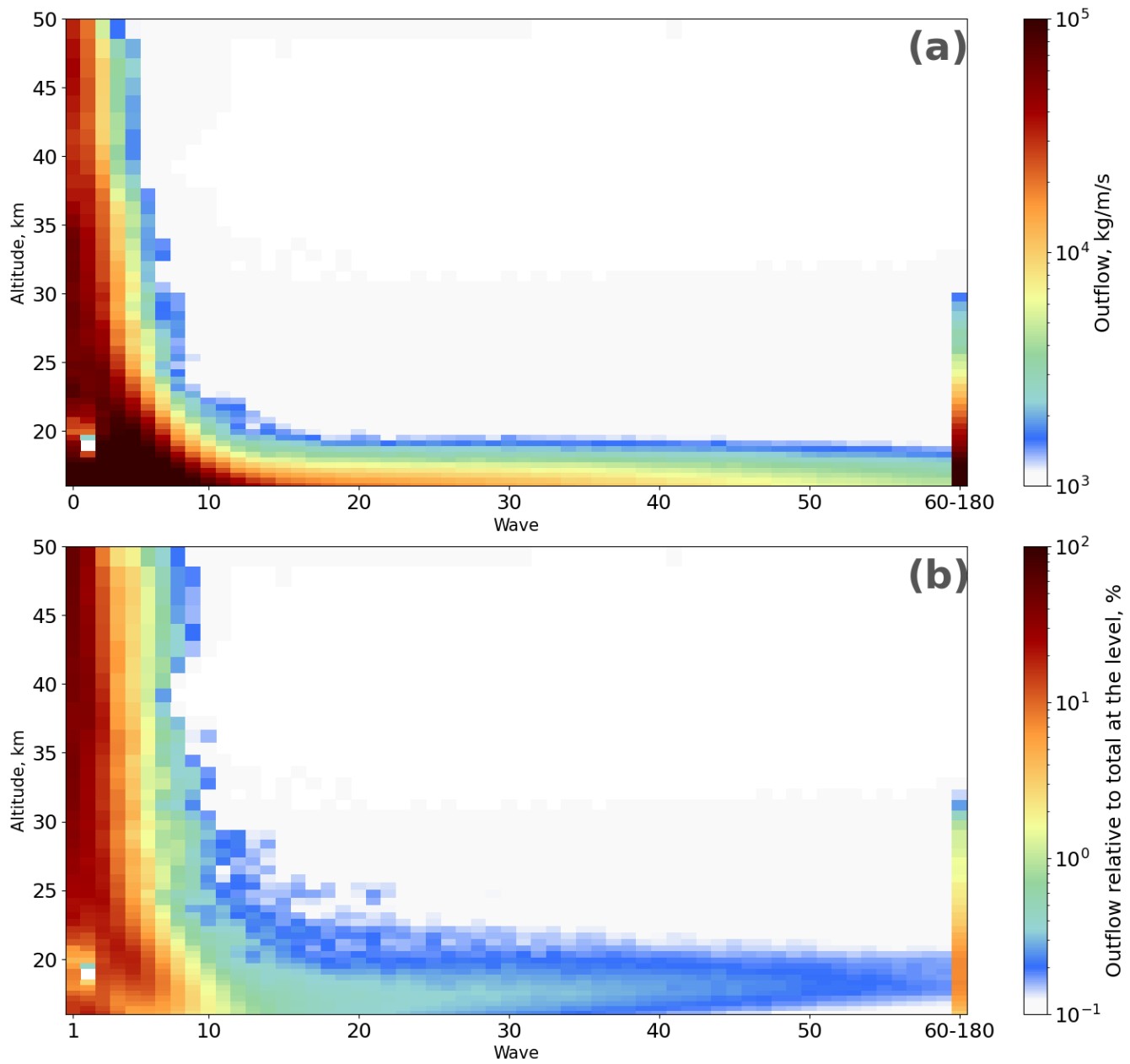

**Figure A1.** (a) 1980-2017 mean vertical profiles of DWCP outflow estimated for different waves from $\nabla \cdot F$ calculations based on ERA5 data. (b) As in (a) but showing the relative contribution (%) of each wave component to the total DWCP outflow at each vertical level.

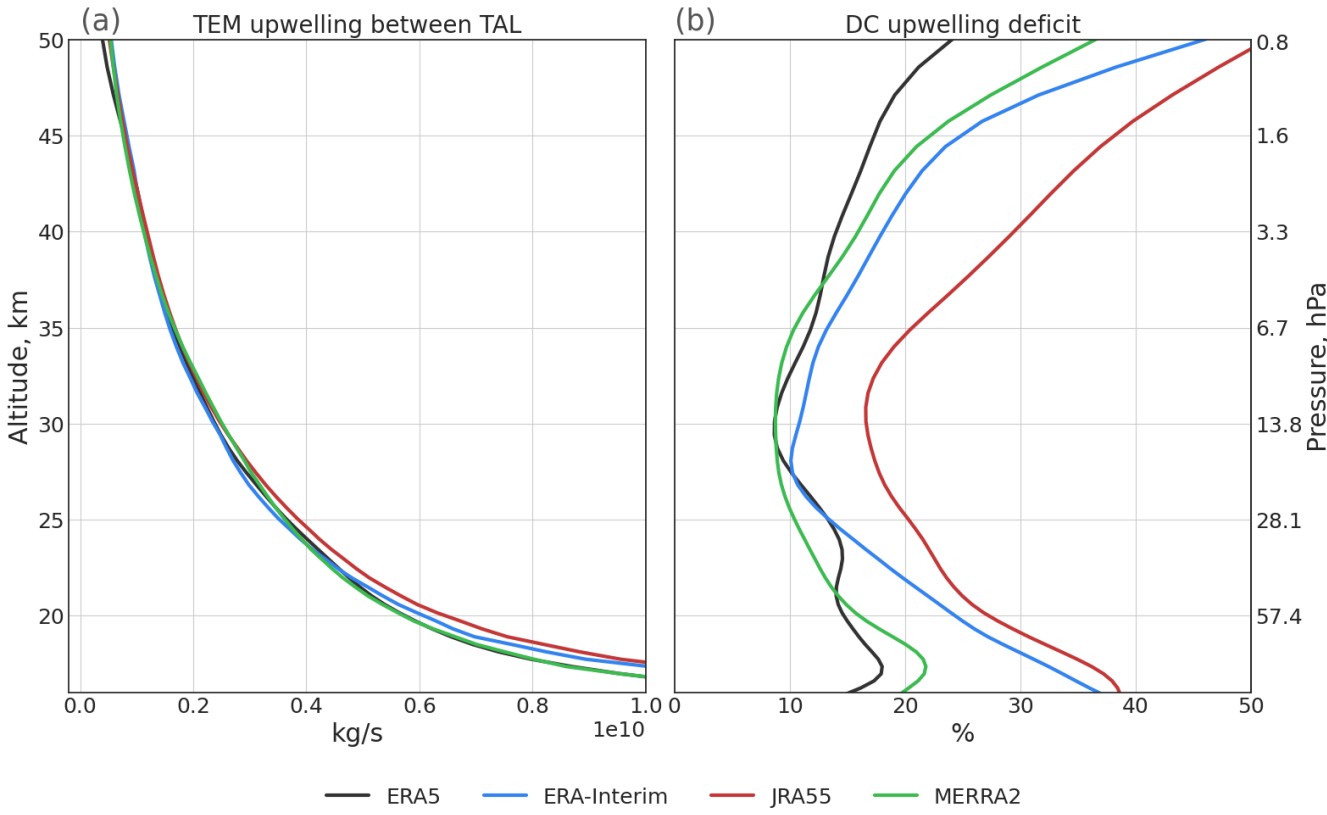

**Figure A2.** (a) 1980-2017 mean profiles of TEM upwelling from ERA5 (black), Era-Interim (blue), JRA55 (red), and MERRA2 (green) reanalyses. (b) The mean upwelling deficit in the downward control calculation (in percent), estimated as the relative difference between TEM and downward control upwelling. This upwelling deficit corresponds to the missing wave drag in the downward calculation (see text). The deficit lines were smoothed with a Gaussian filter. The black line for ERA5 upwelling in (a) in the lower stratosphere mostly overlaps with the green MERRA2 line.

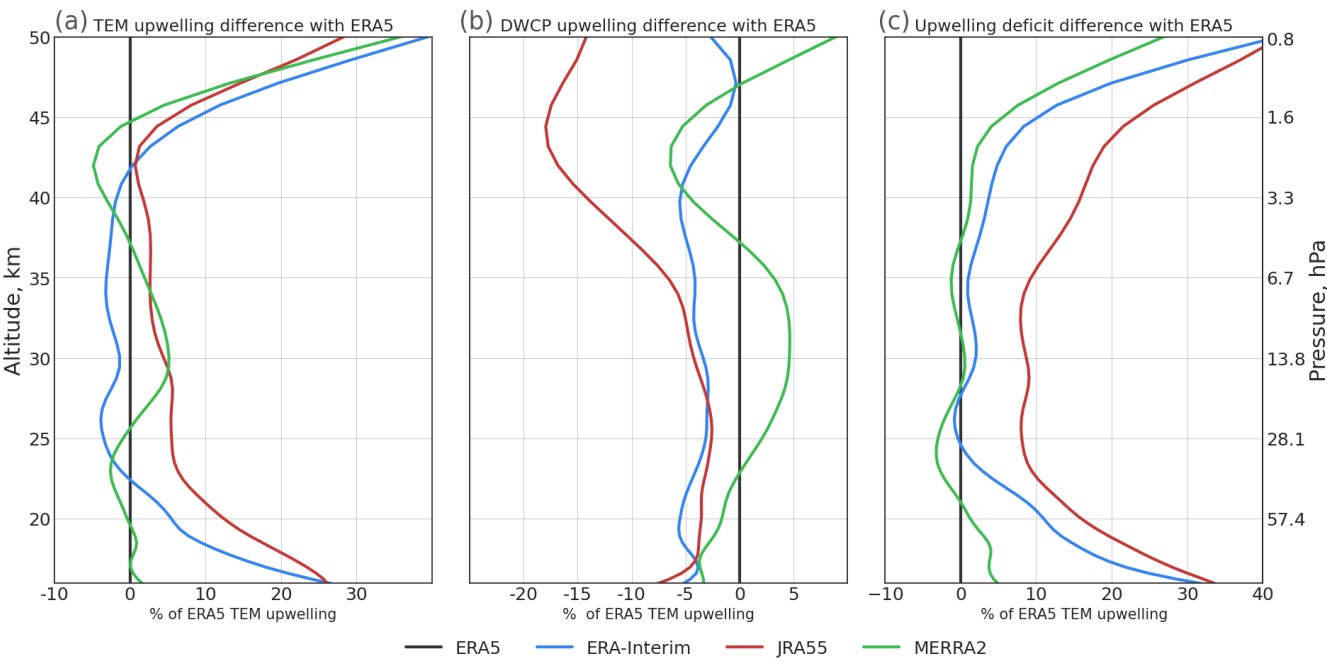

**Figure A3.** Reanalyses upwelling inter-comparison with ERA5 as base, for TEM upwelling (a), DWCP upwelling (b), and upwelling deficit (c)

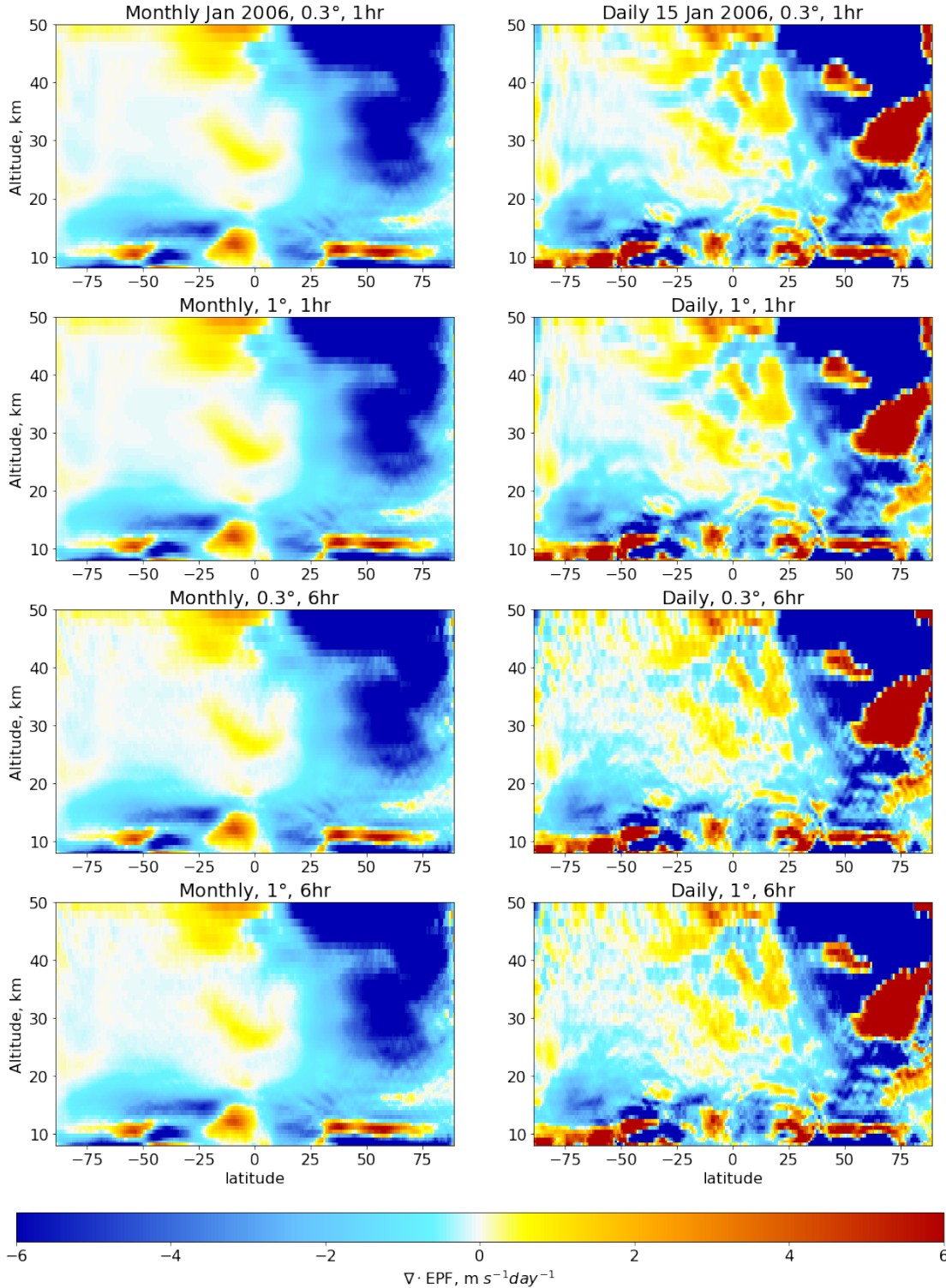

**Figure A4.** Effect of reduction of spatial (from $0.3°$ to $1°$) and temporal (from 1 to 6 hours) resolution on $\nabla\cdot$EPF calculations

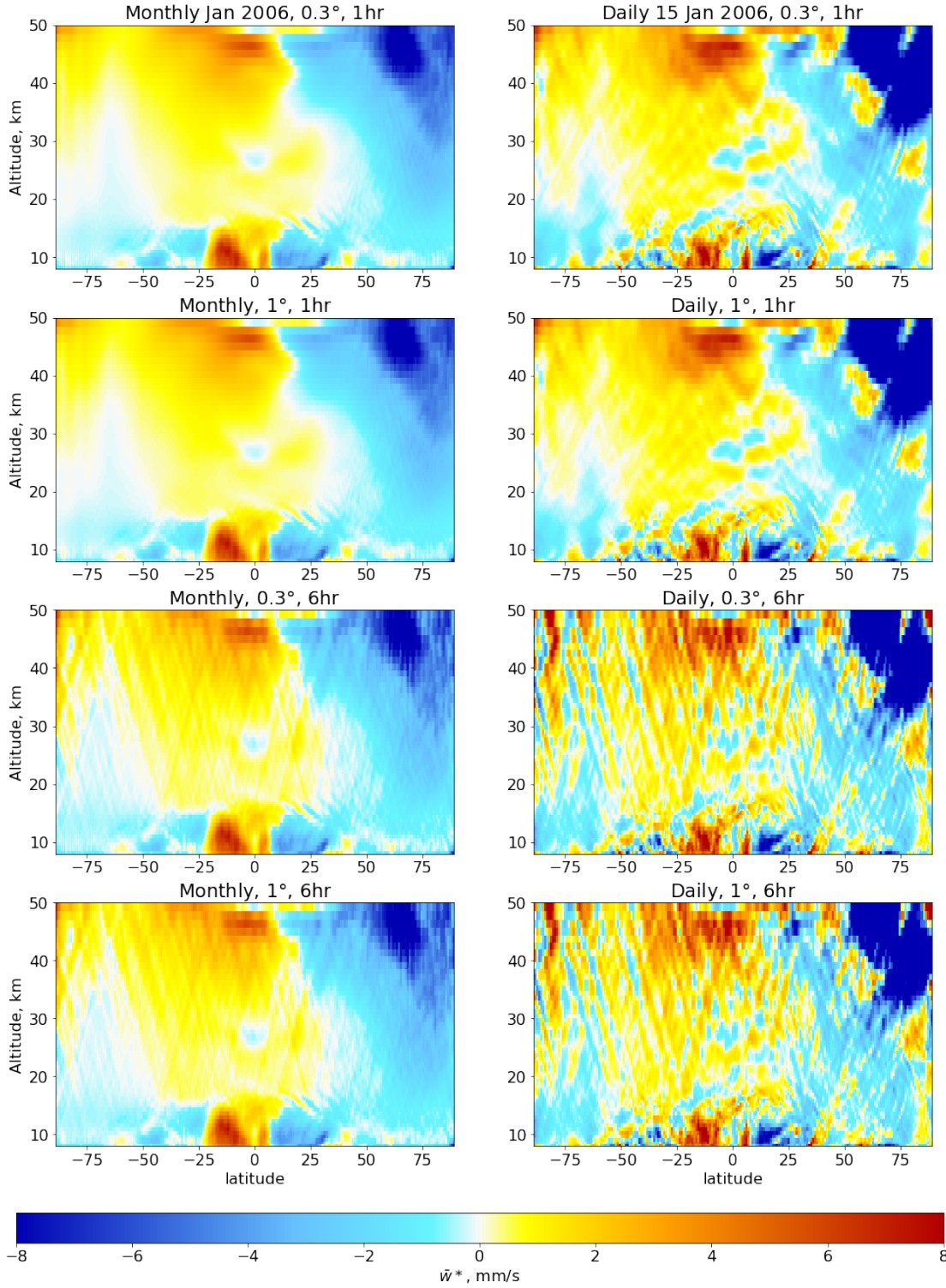

**Figure A5.** Effect of reduction of spatial (from 0.3° to 1°) and temporal (from 1 to 6 hours) resolution on $\bar{w}^*$ calculations