# Peer review of "A dynamics based separation of deep and shallow stratospheric circulation branches"

_EGUsphere, 2024_

## Referee Comment (RC2)

**Review of egusphere-2024-4088: "A dynamical separation of deep and shallow branches in the stratospheric circulation" by Baikhadzhaev et al.**

The study by Baikhadzhaev et al. aims to investigate the stratospheric residual circulation and its wave forcing, focusing on separating the circulation branches. More specifically, they focus on defining the deep and the shallow circulation branch of the residual circulation based on the wave driving and also want to assess changes over decades. These questions are addressed by utilizing reanalysis data, with a focus on ERA5 data along with diagnostics for the residual circulation. They use the TEM framework along with the EP flux divergence to analyze the residual circulation and the wave drag which affects the circulation. They also use the downward control principle to determine the effect of wave dissipation to the residual circulation. A separation between large-scale waves and smaller scale waves is applied to study their contribution to the driving of the lower and the upper part of the stratospheric circulation. Defining upwelling and outflow across the so called turn around latitudes they study the variability and possible trends in the stratospheric circulation as well as they define the level of separation between the two branches. This separation level is found on average at 43 hPa but shows strong seasonal variability, influenced by wave activity and drag. Possible trends in the separation level are found to be not statistically significant, but the level seems to be relatively consistent across different reanalysis data sets. The separation in a deep and shallow branch is proposed based on the wave driving with the deep branch associated with the large scale waves and the shallow branch more related with the smaller scale waves, while the contributions of gravity waves may be underestimated due to the lack of resolution.

The authors aim to improve our understanding of the transport in the stratosphere. This is an important topic since the transport determines the distribution of trace species in the stratosphere which in turn affects the chemistry and radiation in the stratosphere and ultimately can affect processes in the troposphere. Since there is still a lot of uncertainties in how the circulation might change with climate change, further analyses are required in this field and this study can be a valuable contribution and it definitely fits well into Atmospheric Chemistry and Physics. The written language of the manuscript is acceptable. Sometimes the line of thought is hard to follow. The figures are of good quality and support the statements of the text. Section 3 and 4 would profit from some subsections which would separate the various topics better from each other and would increase the readability. In total, I see several points which should be addressed before I would recommend a final publication which from my point of view sum up to major revision. I will lay out my comments in detail below.

**Comments**

1. One major aspect of the paper is the separation of waves into large scale and smaller scales waves. However, I did not really understand why the authors limit themselves to only two categories separated at between wavenumbers 3 and 4. In particular, I found discussions could have been sharpened when a further separation would be used, either by at least separating the synoptic scale from the meso scale. One could go as far and do really a wave number separation to better assess the contributions of the individual wave numbers (at least for the planetary and synoptic scale) to the wave driving of the stratosphere (like in Fig. A1). In particular, the discussions in section 4 would profit substantially from a further

splitting.

2. Although the theoretical background is provided I am still wondering why the authors decided to include the TEM and the downward control principle in their analysis? This could be presented in more detail. In particular, what is the benefit of using both approaches? Is there a complementary aspect or is it just to see whether both diagnostics will result in the same answer? I also found it often difficult to distinguish whether the authors discuss the TEM or the DWCP results throughout the text. Maybe it would be good to separate the analysis more clearly and bring them together in the discussion (see also point 4).

3. I have some issues with the explanation why waves with wave number 180 are considered in this study (lines 131-134). It is stated that a wave can be resolved by two data points. That is against any viable source which I know related to resolution in atmospheric modeling. To resolve an atmospheric phenomenon in a numerical model usually several grid points are required, the number varies between 4 and 8 (one source related to data from ECMWF would be: https://www.ecmwf.int/sites/default/files/elibrary/2013/17358-effective-spectral-resolution-ecmwf-atmospheric-forecast-models.pdf). Also since the reanalysis data in this study has been coarse grained, I wonder how this translates into the resolvable wavenumber. So I wonder how the definition of resolution used in this study fits to the effective resolution commonly used in atmospheric modeling?

4. Section 3 contains several topics which are discussed. It starts with the residual circulation and the EP flux divergence, goes over the upwelling and outflow at the TAL and adds an analysis based on statistical and mechanistic methods about the vertical contribution of the waves driving the circulation. And finally a discussion using all reanalysis data comes along with Fig. 4. I would highly recommend to split this section into more subsections with meaningful headings to provide more structure and increase the readability. I also think that this will definitely help to present the results in a more obvious way.

   In Fig. 2 around line 194 the difference between the TEM and DWCP derived vertical velocity is discussed and is mainly attributed to the parameterized gravity waves. Maybe I got something wrong here, but the differences in question are about 50 % (0.1 m/s vs 0.15 m/s) which is in my opinion quite large to be attributed to the wave drag from parameterized GW. Can you provide more evidence here that the effect of the parameterized gravity waves is most responsible for the difference?

   The discussion around Fig. 3 starting from line 208 is hard to follow but I think is quite essential since the authors use their findings in Fig 3 to determine the separation level. Maybe it would help if is first of all better motivated why a statistical and a mechanistic approach is used here and if they are discussed first clearly separated from each other in individual paragraphs.

   It is also stated in this section 3 that the separation level for ERA5 is mostly higher than for all other reanalysis. This is attributed to the increased potential of ERA5 to resolve gravity waves. Since ERA5 also has the finest vertical resolution in the stratosphere, can you rule out that this result is not simply an effect of the finer grid spacing?

5. In section 4 two topics are discussed (the upwelling and the trends) and I would also suggest to separate them here more visually by introducing subsections.

   The upwelling is also mainly discussed for ERA5 while the trends are immediately discussed for all 4 reanalysis data sets. To my eyes this looks a bit like an inconsistency and the authors do not really explain themselves why they sometimes use all reanalysis data sets and why they sometimes only focus only on ERA5 within a section. Generally, I would appreciate it if all results would be discussed for all reanalysis data sets in the manuscript consistently.

6. Based on my comment 4 about the impact of parameterized GW drag, in section 5, line 404ff the C-like shape of the upwelling deficit is again related to the parameterized GWs. Can the authors support their claim here with additional reasoning or data?

7. I find the comparison shown in Figure A5 quite interesting, in particular the effect of 1h to 6h. The 6h data seems to include spurious effects in w*, which are more prominent in the daily but which are even seen in the monthly mean data. Do the authors know the source of these patterns? Can they have a lasting impact on the analysis?

Technical comments

- Line 50: Do you mean: "stratospheric circulation is expected to accelerate" ?
- Section 2.1: The differences in horizontal resolutions are addressed but not in the vertical. Can this be included? In particular, is the data used on model or pressure levels? What are the differences in vertical grid spacing between the various reanalysis data sets in the stratosphere?
- Line 150: (e.g. (Abalos et al., 2015)) → (e.g., Abalos et al., 2015)
- Line 163: condition(Vallis, 2006) → condition (Vallis, 2006)
- Line 189: vbarstar
- Line 193: DWCP Eq. 5a →  DWCP (see Eq. 5a)
- Line 210: negtive → negative
- Line 283: banches → branches
- Fig A2a: TEM upwelling betwen TAL →  TEM upwelling between TAL

---

## Author Comment (AC2)

We thank the reviewer for the thorough reading of the manuscript and the insightful comments, which clearly helped to improve the manuscript. We addressed all points in the revised version as described below. Reviewer comments are in normal font, answers in italics. A summary of the main changes in the revised version is given in the bullet point list here (with more details given in the replies to the specific comments below).

- The title of the manuscript has been revised to more accurately reflect its content, including the requested modifications.

- In response to Reviewer 2's suggestion, we refined the scale-based separation to distinguish between planetary, synoptic-scale, and mesoscale waves. The separation is now presented in sect. 2.2, and discussed with the new Fig. 3 in sect. 3.

- A decomposition into more physical atmospheric wave phenomena was requested by the reviewer 1. In response, the relation between wave scales and physical wave types is discussed at the end of sect. 2.2: planetary waves are associated with quasi-stationary Rossby waves, synoptic-scale waves represent a mixture between Rossby and gravity waves, and mesoscale waves are associated with gravity waves.

- As requested by Reviewer 3, we conducted additional analysis of upwelling trends to examine whether separating the ozone recovery period from the post-ozone recovery period improves the statistical significance of the trends.

- More information from reanalyses other than ERA5 was added to the sect. 3 and 4 of the manuscript as requested by Reviewer 2

- As requested by Reviewer 2 and 3 rewriting and restructuring of several text parts throughout the entire manuscript was done to improve clarity and readability eg. sect. 2.2, sect. 3 and sect. 4.

- The comparison of resampled ERA5 data with original fine resolution ERA5 data was expanded (see Sect. 5), as requested by reviewer 2.

This paper uses reanalysis data primarily ERA-5 data to determine if stratospheric circulation branches may be defined based on wave-forcing. Three science questions are posed: (1) Can different branches in the stratospheric circulation be dynamically defined based on the wave driving? (2) Did different branches of the stratospheric circulation change differently over the past decades? (3) How robust is the representation of the circulation structure and its changes in different reanalyses? Results identify two robust branches: a deep branch predominantly driven by waves with wavenumbers 1 to 3 and a shallow branch predominantly driven by waves with wavenumbers 4 to 180. Results also find a long-term trend in these branches.

My main issue is that the paper performed the calculations based on a definition of atmospheric waves solely in terms of wavenumber. This paper quantifies the contributions of atmospheric waves on these deep and shallow circulation branches. The waves are separated based on their wavenumbers. There's a category of waves comprising those with wavenumbers 1 to 3 and another for waves with wavenumber 4 to 180. There is no mention of phase and periodicities. Taking this approach, the paper doesn't properly decompose the dynamical variables into real physical atmospheric wave phenomena. Hence, there is questionable physical meaning/significance to the results.

The deep and shallow circulation branches that are reported in the introduction seem to allude to winter circulation. Hence, a proper analysis should've focused on real physical planetary-scale waves (e.g. wave-1 VS wave-2 and/or Stationary vs travelling waves) that are known to occur when these branches form. The analysis should have involved processing daily reanalysis data. The paper actually doesn't clearly state how the problematic decompositions are made. Some sections indicate the decomposition was done on monthly-means which isn't sufficient. Either way, defining waves solely in terms of wavenumber was a major flaw. Consequently, the analysis of long-term changes are rendered wrong. Owing to this, I cannot recommend this paper for publication. I suggest the authors take guidance from papers like Yasui et al [2021] and Koval et al [2023] on how to properly quantify the role of atmospheric waves on any middle atmospheric phenomena. You'll see all of them properly acknowledging different kinds of physical planetary-scale waves (e.g. Rossby waves, kelvin waves, etc).

*We thank the reviewer for the thorough reading and evaluation of the manuscript and the critical comments.*

*In principle, wave types can be distinguished using a variety of criteria, including spatial and temporal scales, underlying restoring mechanism, or variable of interest. There is no universally correct classification scheme; the appropriate choice depends on the specific context and the aspects of wave behavior one aims to interpret. The analysis in this study is performed on a data record of 38 years length and for four different reanalysis data sets. This requires a diagnostic approach which keeps the computational effort of the analysis affordable and motivates our decision for a solely-longitudinal scale separation. In the stratosphere, such an approach is well justified and commonly used (e.g. Abalos et al., 2024; McLandress and Shepherd, 2009; Kim et al., 2016). We agree that in the mesosphere and at even higher altitudes, issues can arise when applying simple scale separation as the reviewer has pointed out. However, the main focus of our paper is on the region near the separation level in the lower stratosphere (below approximately 30 km). Here, Kelvin waves are typically confined to within approximately $\pm 10^o$ of the equator, away from the turn-around latitudes. As the tropical upwelling of the stratospheric circulation at these altitudes is mainly driven by wave drag around the turn-around latitudes in the subtropics, Kelvin waves therefore have a very limited impact on the circulation. Furthermore,*

*some insight into the physical wave types can be inferred based on their spatial scales: planetary waves (wavenumbers 1–3) are most likely stationary or quasi-stationary Rossby waves; synoptic-scale waves (wavenumbers 4–20) are dominated by Rossby waves with increasing secondary contributions from gravity waves for higher wavenumbers; and mesoscale waves (wavenumbers greater than 20) are likely dominated by gravity waves. A related discussion linking wave scales to physical wave types has been incorporated in the revised version of the manuscript at the end of Sect. 2.2 and is further elaborated in Sect. 5.*

*We agree that a physical separation of the waves would be interesting, but this would be another, complementary study. Using reanalysis data, which do not have an inherent separation of wave modes, this would require a space-time decomposition which is costly. Hence, a physical separation is immensely complex, would need thorough validation and thus is unfortunately out of scope of the paper.*

*We further would like to clarify that the decomposition was performed on instantaneous, regularly sampled global reanalysis data (6-hourly basis, even 1-hourly for the high-resolution ERA5 test case). All averaging in the paper is performed after evaluation of these snapshots. Thus, the effects of all wave modes on the EP-flux are kept and no information is lost by averaging. To avoid confusion, this is described in a much clearer way now at the end of the methods Sect. 2.2.*

Other issues:

- This paper is centered on the deep and shallow circulation branches that are centered on a TAL between 20 N/S and 40 N/S. In the introduction, the paper acknowledges previous studies identifying shallow and deep branches in the stratospheric circulation. However, the science questions made it seem as though this paper aimed to find other very distinct kinds of circulation. The science questions need to be re-written to clarify that this paper still focuses on these deep and shallow circulation branches.

  *We thank the reviewer for pointing out this potential for confusion regarding the research questions. We have clarified the related text to present the scientific questions more clearly, specifically emphasizing that the deep and shallow circulation branches are the focus of this paper. The revised questions can be found at the end of the introduction in Sect. 1.*

- Add a paragraph at the end of section 2.2 that explains how each science question can be answered using these parameters (and any accompanying methodology)? This will help readers immediately know what to look out for.

*The paragraph is added at the end of Sect. 2.2. To address the research questions outlined in the Sect. 1, a Fourier transformation is applied to 6-hourly snapshots of the fluctuation components, and $\nabla \cdot F(s)$ is calculated (see Eq. 4). After that, the monthly mean is computed from the $\nabla \cdot F(s)$ snapshots. Utilizing the DWCP framework (Eq. 5), the monthly mean $\nabla \cdot F(s)$ related to specific wavenumbers is used to reconstruct the circulation driven by individual waves or sets of waves. Subsequently, the associated upwelling and outflow are computed from the reconstructed circulation using Eq. 7, and 8. The analysis is then conducted on the resulting upwelling and outflow fields.*

- The authors need to introduce figures first before reporting on results found in the figures.

  *We agree that at several points the figures could be introduced and described in a clearer way to make it easier to the readers to follow the line of argumentation. We carefully worked through the entire manuscript to make sure that figures are always explained clearly.*

- The authors need to make use of the mathematical symbols of terms (that they already introduced). Some sections are too wordy because instead of using the short-cut symbols, the authors still write the complete name/phrase of the parameter.

  *Thanks for the comment. It is always a balance between conciseness and readability when using mathematical symbols. To maintain readability also for readers who are not familiar with the mathematical language of the TEM formalism and stratospheric circulation we find it beneficial not to use mathematical symbols too excessively. Nevertheless, we followed the reviewer's suggestion partly and carefully worked through the entire manuscript to shorten the text by use of symbols wherever we found that reasonable.*

References:

Abalos, M., Randel, W. J., Garcia, R. R. (2024), The Dominant Role of the Summer Hemisphere in Subtropical Lower Stratospheric Wave Drag Trends, J. Geophys. Res., https://doi.org/10.1029/2023GL105827

Kim, J., Randel, W. J., Birner, T., Abalos, M. (2016), Spectrum of Wave Forcing Associated with the Annual Cycle of Upwelling at the Tropical Tropopause, J. Atmos. Sci., https://doi.org/10.1175/JAS-D-15-0096.1

Koval, A. V., Toptunova, O. N., Motsakov, M. A., Didenko, K. A., Ermakova, T. S., Gavrilov, N. M., & Rozanov, E. V. (2023). Numerical modelling of relative contribution of planetary waves to the atmospheric circulation. *Atmospheric Chemistry and Physics*, *23*(7), 4105-4114.

McLandress, C., and Shepherd, T., G. (2009), Simulated Anthropogenic Changes in the Brewer–Dobson Circulation, Including Its Extension to High Latitudes, J. Atmos. Sci., https://doi.org/10.1175/2008JCLI2679.1

Yasui, R., Sato, K., & Miyoshi, Y. (2021). Roles of Rossby waves, Rossby–gravity waves, and gravity waves generated in the middle atmosphere for interhemispheric coupling. *Journal of the Atmospheric Sciences, 78*(12), 3867-3888.

---

## Author Comment (AC3)

We thank the reviewer for the thorough reading of the manuscript and the insightful comments, which clearly helped to improve the manuscript. We addressed all points in the revised version as described below. Reviewer comments are in normal font, answers in italics. A summary of the main changes in the revised version is given in the bullet point list here (with more details given in the replies to the specific comments below).

- The title of the manuscript has been revised to more accurately reflect its content, including the requested modifications.

- In response to Reviewer 2's suggestion, we refined the scale-based separation to distinguish between planetary, synoptic-scale, and mesoscale waves. The separation is now presented in sect. 2.2, and discussed with the new Fig. 3 in sect. 3.

- A decomposition into more physical atmospheric wave phenomena was requested by the reviewer 1. In response, the relation between wave scales and physical wave types is discussed at the end of sect. 2.2: planetary waves are associated with quasi-stationary Rossby waves, synoptic-scale waves represent a mixture between Rossby and gravity waves, and mesoscale waves are associated with gravity waves.

- As requested by Reviewer 3, we conducted additional analysis of upwelling trends to examine whether separating the ozone recovery period from the post-ozone recovery period improves the statistical significance of the trends.

- More information from reanalyses other than ERA5 was added to the sect. 3 and 4 of the manuscript as requested by Reviewer 2

- As requested by Reviewer 2 and 3 rewriting and restructuring of several text parts throughout the entire manuscript was done to improve clarity and readability eg. sect. 2.2, sect. 3 and sect. 4.

- The comparison of resampled ERA5 data with original fine resolution ERA5 data was expanded (see Sect. 5), as requested by reviewer 2.

**General comment**

The study by Baikhadzhaev et al. aims to investigate the stratospheric residual circulation and its wave forcing, focusing on separating the circulation branches. More specifically, they focus on defining the deep and the shallow circulation branch of the residual circulation based on the wave driving and also want to assess changes over decades. These questions are addressed by utilizing reanalysis data, with a focus on ERA5 data along with diagnostics for the residual circulation. They use the TEM framework along with the EP flux divergence to analyze the residual circulation and the wave drag which affects the circulation. They also use the downward control principle to determine the effect of wave

dissipation to the residual circulation. A separation between large-scale waves and smaller scale waves is applied to study their contribution to the driving of the lower and the upper part of the stratospheric circulation. Defining upwelling and outflow across the so called turn around latitudes they study the variability and possible trends in the stratospheric circulation as well as they define the level of separation between the two branches. This separation level is found on average at 43 hPa but shows strong seasonal variability, influenced by wave activity and drag. Possible trends in the separation level are found to be not statistically significant, but the level seems to be relatively consistent across different reanalysis data sets. The separation in a deep and shallow branch is proposed based on the wave driving with the deep branch associated with the large scale waves and the shallow branch more related with the smaller scale waves, while the contributions of gravity waves may be underestimated due to the lack of resolution.

The authors aim to improve our understanding of the transport in the stratosphere. This is an important topic since the transport determines the distribution of trace species in the stratosphere which in turn affects the chemistry and radiation in the stratosphere and ultimately can affect processes in the troposphere. Since there is still a lot of uncertainties in how the circulation might change with climate change, further analyses are required in this field and this study can be a valuable contribution and it definitely fits well into Atmospheric Chemistry and Physics. The written language of the manuscript is acceptable. Sometimes the line of thought is hard to follow. The figures are of good quality and support the statements of the text. Section 3 and 4 would profit from some subsections which would separate the various topics better from each other and would increase the readability. In total, I see several points which should be addressed before I would recommend a final publication which from my point of view sum up to major revision. I will lay out my comments in detail below.

*We thank the reviewer for the overall positive evaluation of the manuscript. We see the criticism that the text needs improvement. Therefore, we carefully worked on improving the entire manuscript text and worked on the wording as well as on the structure, in order to enhance the clarity of the presentation. Also, we separated Sections 3 and 4 into further subsections as suggested.*

**Detailed comments**

1. One major aspect of the paper is the separation of waves into large scale and smaller scales waves. However, I did not really understand why the authors limit themselves to only two categories separated at between wavenumbers 3 and 4. In particular, I found discussions could have been sharpened when a further separation would be used, either by at least separating the synoptic scale from the meso scale. One could go as far and do really a wave number separation to better assess the contributions of the individual wave numbers (at least for the planetary and synoptic scale) to the wave driving of the stratosphere (like in

Fig. A1). In particular, the discussions in section 4 would profit substantially from a further splitting.

*We thank the reviewer for the good suggestion to present the individual wave contributions more clearly to motivate the separation between wave numbers 3 and 4. We had analysed the individual wave contributions already before but had tried to condense the presented information. We see now that this likely causes more confusion and that it would be helpful to present more details. Hence, at the end of Sect. 2.2 we split atmospheric waves into planetary, synoptic, and mesoscale based on their scale. Upwelling and outflow from these wave types is presented in the new Fig. 3, a more detailed discussion of the wave types contribution to the circulation is now given in Sect. 3.*

2. Although the theoretical background is provided I am still wondering why the authors decided to include the TEM and the downward control principle in their analysis? This could be presented in more detail. In particular, what is the benefit of using both approaches? Is there a complementary aspect or is it just to see whether both diagnostics will result in the same answer? I also found it often difficult to distinguish whether the authors discuss the TEM or the DWCP results throughout the text. Maybe it would be good to separate the analysis more clearly and bring them together in the discussion (see also point 4).

*We understand that the description of the method was not clear enough and that in particular the usage of different diagnostics was not motivated enough. We thank the reviewer for pointing to that. The DWCP is based on the TEM framework, and is used to relate residual circulation velocities to wave drag. On the one hand, the direct TEM approach is conceptually simpler and more frequently used in studies of the stratospheric circulation. Each estimate is subject to distinct uncertainties and limitations, and their comparison provides information on the robustness of our results. We tried to clarify the text throughout the paper, and inserted a few motivating sentences at the beginning of the Sect. 2.2.*

3. I have some issues with the explanation why waves with wave number 180 are considered in this study (lines 131-134). It is stated that a wave can be resolved by two data points. That is against any viable source which I know related to resolution in atmospheric modeling. To resolve an atmospheric phenomenon in a numerical model usually several grid points are required, the number varies between 4 and 8 (one source related to data from ECMWF would be: https://www.ecmwf.int/sites/default/files/elibrary/2013/17358-effective-spectral-resolution-ecmwf-atmospheric-forecast-models.pdf). Also since the reanalysis data in this study has been coarse grained, I wonder how this translates into the resolvable wavenumber. So I wonder how the definition of resolution used in this study fits to the effective resolution commonly used in atmospheric modeling?

*We thank the reviewer for pointing to that potential for misunderstanding. Of course, the maximum zonal wavenumber of 180 for our downsampled ERA5 data will be somewhat different from the effective resolution of ERA5. The original model data has a spectral resolution of TL639 (Hersbach et al., 2020). However, the effective resolution for atmospheric waves will be somewhat worse (e.g., Skamarock, 2004). In order to check whether our downsampled ERA5 data loses a significant portion of the waves resolved in ERA5, we have examined the effect of resolution change from fine resolution ERA5 data with 1200 points to coarser resolution with 360 points along 360 degrees of longitude. This sensitivity calculation has been performed for the full year 2010. We found that resampling led to a 20 % reduction in DWCP upwelling driven by waves 21–180 near the tropopause. However, the total upwelling in this region decreased by only 1.5 %. Additionally, we observed an approximately 1.3 % reduction in total outflow near the separation level. Hence we find that the effect of resampling is notable, yet acceptable for this study. The analysis of the effects of resolution changes on calculated upwelling and outflow is discussed in the discussion Sect. 5.*

4. Section 3 contains several topics which are discussed. It starts with the residual circulation and the EP flux divergence, goes over the upwelling and outflow at the TAL and adds an analysis based on statistical and mechanistic methods about the vertical contribution of the waves driving the circulation. And finally a discussion using all reanalysis data comes along with Fig. 4. I would highly recommend to split this section into more subsections with meaningful headings to provide more structure and increase the readability. I also think that this will definitely help to present the results in a more obvious way. In Fig. 2 around line 194 the difference between the TEM and DWCP derived vertical velocity is discussed and is mainly attributed to the parameterized gravity waves. Maybe I got something wrong here, but the differences in question are about 50 % (0.1 m/s vs 0.15 m/s) which is in my opinion quite large to be attributed to the wave drag from parameterized GW. Can you provide more evidence here that the effect of the parameterized gravity waves is most responsible for the difference? The discussion around Fig. 3 starting from line 208 is hard to follow but I think is quite essential since the authors use their findings in Fig 3 to determine the separation level. Maybe it would help if is first of all better motivated why a statistical and a mechanistic approach is used here and if they are discussed first clearly separated from each other in individual paragraphs. It is also stated in this section 3 that the separation level for ERA5 is mostly higher than for all other reanalysis. This is attributed to the increased potential of ERA5 to resolve gravity waves. Since ERA5 also has the finest vertical resolution in the stratosphere, can you rule out that this result is not simply an effect of the finer grid spacing?

*Following advice from the reviewer the section 3 was split into three parts: 1) General structure of the stratospheric circulation 2) Separation of the deep and shallow branches by wave forcing based on climatological data, 3) Level of*

the separation between shallow and deep branches.

Indeed, only about 2/3 of the total TEM upwelling at 70hPa in Fig. 2 are reconstructed with the DWCP method. This number seems high, but is in good agreement with other studies based on other models. For instance Butchart et al (2011) found also that about 1/3 of the upwelling across 70hPa in climate models is related to parameterized waves (their Fig. 10).

Regarding Fig. 3, we acknowledge that the description of Fig. 3 was not sufficiently clear and thank the reviewer for pointing this out. To improve clarity and ensure a more logical flow, panels a and b have been swapped with panels b and c, and the corresponding figure description has been restructured accordingly.

Vertical resolution and the model's ability to resolve gravity waves are highly intertwined eg. Watanabe et al. (2015), where it was found that gravity wave momentum flux is decreasing with finer vertical resolution.

5. In section 4 two topics are discussed (the upwelling and the trends) and I would also suggest to separate them here more visually by introducing subsections. The upwelling is also mainly discussed for ERA5 while the trends are immediately discussed for all 4 reanalysis data sets. To my eyes this looks a bit like an inconsistency and the authors do not really explain themselves why they sometimes use all reanalysis data sets and why they sometimes only focus only on ERA5 within a section. Generally, I would appreciate it if all results would be discussed for all reanalysis data sets in the manuscript consistently.

*Following the reviewer's suggestion, section 4 was split into two subsections: 1) Variability of the circulation, 2) Trends. Additionally, more results from reanalyses other than ERA5 were incorporated into Sections 3 and 4, and Table 2 was expanded to include corresponding data from these additional reanalyses.*

6. Based on my comment 4 about the impact of parameterized GW drag, in section 5, line 404ff the C-like shape of the upwelling deficit is again related to the parameterized GWs. Can the authors support their claim here with additional reasoning or data?

*We thank the reviewer for this question. As noted in our response to Comment 4, an upwelling deficit of about 30 % at 70 hPa due to the absence of parameterized wave drag is not uncommon and is consistent with previous studies. The 'C'-shaped structure of the upwelling deficit can be interpreted as follows: the enhanced deficit at upper levels likely reflects the dominance of gravity waves, which account for nearly all wave forcing in the mesosphere. The increase of deficit in the lower stratosphere is likely related to the fact that gravity wave drag contributes significantly to the shallow branch of the Brewer–Dobson circulation eg. Eichinger et al. (2020); Diallo et al. (2019). This is briefly discussed now in Sect. 5.*

7. I find the comparison shown in Figure A5 quite interesting, in particular

the effect of 1h to 6h. The 6h data seems to include spurious effects in w*, which are more prominent in the daily but which are even seen in the monthly mean data. Do the authors know the source of these patterns? Can they have a lasting impact on the analysis?

*We thank the reviewer for the question. As noted in our response to comment 3, we conducted an additional comparison between the fine-resolution and coarsened ERA5 data and found that the impact of resampling on the analysis is limited. This conclusion is further supported by the minimal effect of resampling on the Eliassen-Palm flux divergence, as shown in Fig. 1. This new figure shows that the patterns in monthly averaged wave drag are not changing significantly between the fine and coarse resolution data, for the total wave drag as well as for the contributions from planetary and smaller-scale waves. The effects of resampling are discussed in more detail now in Sect. 5.*

Technical comments

- Line 50: Do you mean: "stratospheric circulation is expected to accelerate" ?

  *Yes, the text was corrected.*

- Section 2.1: The differences in horizontal resolutions are addressed but not in the vertical. Can this be included? In particular, is the data used on model or pressure levels? What are the differences in vertical grid spacing between the various reanalysis data sets in the stratosphere?

  *Information about vertical resolution of the reanalyses is now included in sect. 2.1.*

- Line 150: (e.g. (Abalos et al., 2015)) → (e.g., Abalos et al., 2015)

- Line 163: condition(Vallis, 2006) → condition (Vallis, 2006)

- Line 189: vbarstar

- Line 193: DWCP Eq. 5a → DWCP (see Eq. 5a)

- Line 210: negtive → negative

- Line 283: banches → branches

- Fig A2a: TEM upwelling betwen TAL → TEM upwelling between TAL

  *We thank the reviewer for the corrections, the text was corrected according to the technical comments.*

References:
Butchart, N., et al. (2011), Multimodel climate and variability of the stratosphere, J. Geophys. Res., 116, D05102,doi:10.1029/2010JD014995.

Eichinger, R., Garny, H., Šácha, P. *et al.* Effects of missing gravity waves on stratospheric dynamics; part 1: climatology. *Clim Dyn* **54**, 3165–3183 (2020). https://doi.org/10.1007/s00382-020-05166-w

Diallo, M., Konopka, P., Santee, M. L., Müller, R., Tao, M., Walker, K. A., Legras, B., Riese, M., Ern, M., and Ploeger, F.: Structural changes in the shallow and transition branch of the Brewer–Dobson circulation induced by El Niño, Atmos. Chem. Phys., 19, 425–446, https://doi.org/10.5194/acp-19-425-2019, https://www.atmos-chem-phys.net/19/425/2019/, 2019.

Hersbach, H., Bell, B., Berrisford, P., Hirahara, S., Horànyi, A., Muñoz Sabater, J., Nicolas, J., Peubey, C., Radu, R., Schepers, D., Simmons, A., Soci, C., Abdalla, S., Abellan, X., Balsamo, G., Bechtold, P., Biavati, G., Bidlot, J., Bonavita, M., De Chiara, G., Dahlgren, P., Dee, D., Diamantakis, M., Dragani, R., Flemming, J., Forbes, R., Fuentes, M., Geer, A., Haimberger, L., Healy, S., Hogan, R. J., Hólm, E., Janiskovà, M., Keeley, S., Laloyaux, P., Lopez, P., Lupu, C., Radnoti, G., de Rosnay, P., Rozum, I., Vamborg, F., Villaume, S., and Thépaut, J.-N.: The ERA5 global reanalysis, Q. J. R. Meteorol. Soc., 146, 1999–2049, https://doi.org/10.1002/qj.3803, 2020.

Skamarock, W. C.: Evaluating Mesoscale NWP Models Using Kinetic Energy Spectra, Monthly Weather Review, 132, 3019 – 3032,715 https://doi.org/10.1175/MWR2830.1

Watanabe, S., Sato, K., Kawatani, Y., and Takahashi, M.: Vertical resolution dependence of gravity wave momentum flux simulated by an atmospheric general circulation model, Geosci. Model Dev., 8, 1637–1644, https://doi.org/10.5194/gmd-8-1637-2015, 2015.

[Figure]

Figure 1: Eliassen–Palm flux divergence (color coded), together with Eliassen–Palm flux vectors from ERA5 reanalysis for the year 2010. The distributions are shown for the contribution from all resolved waves (left), from planetary waves with wavenumbers 1–3 (middle), and from smaller-scale waves (right).

---

## Author Comment (AC4)

We thank the reviewer for the thorough reading of the manuscript and the insightful comments, which clearly helped to improve the manuscript. We addressed all points in the revised version as described below. Reviewer comments are in normal font, answers in italics. A summary of the main changes in the revised version is given in the bullet point list here (with more details given in the replies to the specific comments below).

- The title of the manuscript has been revised to more accurately reflect its content, including the requested modifications.

- In response to Reviewer 2's suggestion, we refined the scale-based separation to distinguish between planetary, synoptic-scale, and mesoscale waves. The separation is now presented in sect. 2.2, and discussed with the new Fig. 3 in sect. 3.

- A decomposition into more physical atmospheric wave phenomena was requested by the reviewer 1. In response, the relation between wave scales and physical wave types is discussed at the end of sect. 2.2: planetary waves are associated with quasi-stationary Rossby waves, synoptic-scale waves represent a mixture between Rossby and gravity waves, and mesoscale waves are associated with gravity waves.

- As requested by Reviewer 3, we conducted additional analysis of upwelling trends to examine whether separating the ozone recovery period from the post-ozone recovery period improves the statistical significance of the trends.

- More information from reanalyses other than ERA5 was added to the sect. 3 and 4 of the manuscript as requested by Reviewer 2

- As requested by Reviewer 2 and 3 rewriting and restructuring of several text parts throughout the entire manuscript was done to improve clarity and readability eg. sect. 2.2, sect. 3 and sect. 4.

- The comparison of resampled ERA5 data with original fine resolution ERA5 data was expanded (see Sect. 5), as requested by reviewer 2.

Review for 'a dynamical separation of deep and shallow branches in the stratospheric circulation' by Rasul et al.

This work uses 4 reanalysis data to separate the deep and shallow branch of the Brewer-Dobson circulation, concluded that the shallow branch is mainly driven by wave number smaller than 180, and the deep branch is mainly driven by wave number larger than 180. The past trend over 1980-2017 is also presented. Overall, the analysis of this work is clear, and the paper is well written. I recommend accept this work after a minor revision.

*We thank the reviewer for the generally positive evaluation of the manuscript.*

Major comment:

The trend that the authors calculated from all reanalysis data over 1980-2017 are not statistically significant. I think this might because over 1980-2017, the Brewer-Dobson circulation could be divided into two eras with an opposite trend: slower during 1980-2000, and faster during 2000-2017 (Polvani et al., 2018, Fu et al, 2019). I suggest that the authors check the trend over the individual time periods.

*We thank the reviewer for this very helpful suggestion to divide the trend analysis into periods before and after year 2000, hence influenced by either ozone depletion or ozone recovery, respectively. We carried out the suggested separation of trend calculation into the two periods (see Fig. 1 in this reply letter). However, this separation did not lead to higher statistical significance in the calculated trends (see Fig. 1). The discussion of this result has now been included in Sect. 4.2 of the revised manuscript.*

Also, throughout of this paper, there are too much wording like 'maybe', 'appears', which sounds not very scientific and professional, please try to avoid it.

*We agree that the text needed improvement. Therefore, we carefully worked on improving the text throughout the entire manuscript and worked on the wording as well as on the structure, in order to enhance the clarity of the presentation.*

Specific comments:
Line 31, 'two-way mixing': need citations
*Citation Garny et al. (2014) added.*

Line 50 'the stratospheric circulation is expected to increase': 'increase' could be misleading, 'strengthen' is better
*The word increase was replaced with accelerate as suggested by the reviewer 2.*

Line 116 'EP flux': needs a citation here
*Citation Andrews (1987) added.*

Equation 4: a more detailed calculation method should be described here. Do you calculate the EP flux of each wave number, and then calculate the mean value for wave 1-3, and 4-180? Do you need to perform an inverse Fourier transform from the wavenumber-frequency domain to longitude?

*To compute the individual wave contributions, a Fast Fourier Transform (FFT) was applied to each fluctuation component along the longitudinal axis. For example, to obtain $\hat{v}(s)$, the FFT is applied to the zonal array of $v'$, and only the positive wavenumbers are retained. The resulting spectrum $F(s)$ consists of $N/2$ wavenumbers, each representing the contribution of a specific zonal wavenumber, effectively replacing the longitude dimension in the transformed space.*

*A more detailed description of the Eq. 4 is now present in Sect. 2.2.*

Line 133 'nyquist criterion': citation here.
*Citations Nyquist (1928) and Shannon (1949) added.*

Line 172: DeltaT = 86400s. I think to make the calculation balanced; the unit should be 86400 s/day. And there should be a space between the number and the units. Please also check other numbers in this manuscript.
*Fixed.*

Figure 1 caption: Is white contour arrows only in figure 1a, and black contour arrows only in figure 1b-c? Why choose to compare different variables?
*Different variables are shown in the different panels to enable comparing the residual circulation as well as EP flux vectors.*

Line 179 'the wind threshold': is there a relevant citation?
*Citation Charney and Drazin (1961) added.*

Line 185: this sentence is not Grammarly correct.
*Sentence restructured.*

Figure 2: green line: you might want to name it 'TEM residual' instead of 'TEM' to avoid possible confusion.
*We added a few sentences at the beginning of Sect. 2.2 to clarify what TEM refers to in the figure.*

Line 240: 'ERA5 is able to resolve gravity waves better than other reanalysis': need a citation here
*citation Hoffman et al. (2019) added.*

Line 267: 'we expect a higher..': can you explain more about this sentence?
*Figure A2 illustrates a decrease in the upwelling deficit with altitude within the shallow branch region, suggesting that parameterized wave drag contributes substantially to the outflow in this part of the circulation. Since the separation level is defined as the lowest altitude where the outflow driven by planetary waves exceeds that driven by smaller-scale waves, the enhanced contribution of smaller-scale waves—due to the inclusion of parameterized wave drag—is likely to shift the separation level upward. Text in the manuscript is also updated*

Figure 6: it will be helpful to include error bars to the mean value?
*Error bars are added to the Fig.*

320: 'compareed' typo.
*Fixed.*

429: 'we find out the deep branch', the deep branch of the Brewer-Dobson circulation.

*Sentence restructured.*

References:

Polvani, L. M., Abalos, M., Garcia, R., Kinnison, D., & Randel, W. J. (2018). Significant weakening of Brewer-Dobson circulation trends over the 21st century as a consequence of the Montreal Protocol. *Geophysical Research Letters*, 45, 401–409. https://doi.org/10.1002/201

Fu, Q., Solomon, S., Pahlavan, H. A., & Lin, P. (2019). Observed changes in Brewer–Dobson circulation for 1980–2018. *Environmental Research Letters*, *14*(11), 114026.

Garny, H., Birner, T., Bönisch, H., and Bunzel, F.: The effects of mixing on Age of Air, J. Geophys. Res., 119, https://doi.org/10.1002/2013JD021417, 2014.

Andrews, D. G., Holton, J. R., and Leovy, C. B.: Middle Atmosphere Dynamics, Academic Press, San Diego, USA, 1987.

Nyquist, H.: Certain topics in telegraph transmission theory, Transactions of the American Institute of Electrical Engineers, 47, 617–644, https://doi.org/10.1109/T-AIEE.1928.5055024, 1928

Shannon, C. E.: Communication in the Presence of Noise, Proceedings of the IRE, 37, 10–21, https://doi.org/10.1109/JRPROC.1949.232969, 1949.

Charney, J. G. and Drazin, P. G.: Propagation of planetary-scale disturbances from the lower into the upper atmosphere, J. Geophys. Res., 66, 83–109, https://doi.org/10.1029/JZ066i001p00083, http://dx.doi.org/10.1029/JZ066i001p00083, 1961.

Hoffmann, L., Günther, G., Li, D., Stein, O., Wu, X., Griessbach, S., Heng, Y., Konopka, P., Müller, R., Vogel, B., and Wright, J. S.: From ERA-Interim to ERA5: the considerable impact of ECMWF's next-generation reanalysis on Lagrangian transport simulations, Atmos. Chem. Phys., 19, 3097–3124, https://doi.org/10.5194/acp-19-3097-2019, https://www.atmos-chem-phys.net/19/3097/2019/, 2019.

[Figure]

Figure 1: Vertical profiles of upwelling trends constructed from ERA5 data, for periods 1980-1999 (left), and 2000-2017 (right).

---

## Referee Report (RR1)

This revised draft has addressed my issue on clarifying that the paper does indeed focus on these deep and shallow circulation branches. It is clearer now that they wanted to determine the contributions of different atmospheric waves on the structure and variabilities of these branches. As for my concern on the classification of the atmospheric waves:

1.  I mentioned Kelvin waves as an example because I initially thought your paper was interested in finding new circulation branches and not just these mid-latitude branches. Now that it is clear that the branches of interest are these deep and shallow branches, I agree that equatorial Kelvin waves are definitely not relevant to the analysis.
2.  I understand the need to take an approach that is not too computationally heavy. But this approach to acknowledge the importance of waves with wavenumber greater than 4 and even up to wavenumber 180 is still computationally heavy while also not very insightful. Using figure 4 as an example, it is easy to look at the plot and think that wave 3 to 6 clearly induce something that is separate from waves 1 to 2. But then, one still has to ask, what is wave 3? Wave 6? Simply saying that they are mesoscale waves is not enough. My new suggested revision is to perhaps separate the waves into wavenumbers 1 – 2 VS "higher order" waves. From your results, it is clear that wavenumber 1 to 2 clearly drives the deeper branch. I know wavenumbers 3-4 is included in the calculations but again, what are these waves? At higher altitudes, we care about wavenumber 4 planetary-scale waves but I don't think this is recognized in the stratosphere. I also see that "higher order" waves are driving the shallow branch. I'd leave it at that and not over-analyze any specific wavenumber unless you are willing to define that these specific waves are; that is, what's their common phase-speed, propagation of direction, etc.

In summary, the conclusions showing that the deeper branch seems primarily driven by planetary-scale waves while the shallow branch is primarily driven by non-planetary-scales waves is worth publishing. I would just remove any specific investigation on these "higher order" waves unless you are willing to specifically introduce these waves and really prove their Physics or physical significance.

---

## Author Response (AR2)

We thank the editor for his time, and the reviewers for the thorough reading of the manuscript and the insightful comments.

We agree that the suggestion of the reviewer 3 to analyze intermediate waves in more depth could provide valuable additional information. However addressing this issues requires considerable effort, and is beyond the scope of this paper. We see such additional analysis as an opportunity for future research. To reflect this, we have revised the final sentence of the discussion section to highlight the need for a more physically based separation of intermediate-scale waves (wavenumbers 3 and 4) as a promising direction for future investigation.